# Minimally dependent activity subspaces for working memory and motor preparation in the lateral prefrontal cortex

Cheng Tang[1], Roger Herikstad[2], Aishwarya Parthasarathy[1], Camilo Libedinsky[1,2,3†]*, Shih-Cheng Yen[2,4†]*

[1]Institute of Molecular and Cell Biology, A*STAR, Singapore, Singapore; [2]The N1 Institute for Health, National University of Singapore (NUS), Singapore, Singapore; [3]Department of Psychology, NUS, Singapore, Singapore; [4]Innovation and Design Programme, Faculty of Engineering, NUS, Singapore, Singapore

**Abstract** The lateral prefrontal cortex is involved in the integration of multiple types of information, including working memory and motor preparation. However, it is not known how downstream regions can extract one type of information without interference from the others present in the network. Here, we show that the lateral prefrontal cortex of non-human primates contains two minimally dependent low-dimensional subspaces: one that encodes working memory information, and another that encodes motor preparation information. These subspaces capture all the information about the target in the delay periods, and the information in both subspaces is reduced in error trials. A single population of neurons with mixed selectivity forms both subspaces, but the information is kept largely independent from each other. A bump attractor model with divisive normalization replicates the properties of the neural data. These results provide new insights into neural processing in prefrontal regions.

*For correspondence:
camilo@nus.edu.sg (CL);
shihcheng@nus.edu.sg (S-CY)

†These authors contributed equally to this work

Competing interests: The authors declare that no competing interests exist.

## Introduction

Complex flexible behaviors require the integration of multiple types of information, including information about sensory properties, task rules, items held in memory, items being attended, actions being planned, and rewards being expected, among others. A large proportion of neurons in the lateral prefrontal cortex (LPFC) encode a mixture of two or more of these types of information (*Rigotti et al., 2013*; *Parthasarathy et al., 2017*; *Masse et al., 2019*; *van Ede et al., 2019*; *Marcos et al., 2019*). This mixed selectivity endows the LPFC with a high-dimensional representational space (*Rigotti et al., 2013*), but it also presents the challenge of understanding how downstream regions that receive mixed-selective input from the LPFC can read out meaningful information. One possible solution would be to have multiple low-dimensional information subspaces, embedded within the high-dimensional state space of LPFC, which could enable the independent readout of different types of information with minimal interference from changes of information in other subspaces (*Remington et al., 2018*; *Parthasarathy, 2019*; *Semedo et al., 2019*; *Mante et al., 2013*; *Wolff et al., 2019*; *Druckmann and Chklovskii, 2012*). Information subspaces have been identified in the medial frontal cortex (*Wang et al., 2018*), lateral prefrontal cortex (*Parthasarathy, 2019*), early visual areas (*Semedo et al., 2019*), and motor cortex (*Kaufman et al., 2014*; *Elsayed et al., 2016*). However, no studies to date have explicitly tested whether information about two separate cognitive processes can be simultaneously encoded in subspaces within a single biological neural network. Here, we demonstrate the existence of two minimally dependent

information subspaces in the LPFC network: (1) a working memory subspace in which target information emerged in Delay 1, and was maintained till the end of Delay 2; and (2) a motor preparation subspace in which information emerged only in Delay 2 after the presentation of the distractor, possibly due to the initiation of saccade preparation after the last sensory cue that reliably predicted the timing of the Go cue (i.e. the offset of the distractor). Both subspaces exhibited behavioral relevance with significantly decreased information in error trials only in the subspace, and not in the null space. Interestingly, we found a reduction in information in the memory subspace when information in the movement preparation subspace emerged. At the same time, we found that the average firing rate of the neurons across the population remained unchanged. This suggested that a normalization mechanism could have been acting on the population activity (*Ruff and Cohen, 2017*; *Duong et al., 2019*). We subsequently found that a bump attractor model (*Compte et al., 2000*) with divisive normalization allowed us to replicate the observed neurophysiological properties. We believe these results provide insights into the neural mechanisms of cognitive flexibility and cognitive capacity.

## Results

We measured LPFC activity from two monkeys while they performed a delayed saccade task with an intervening distractor. Briefly, the monkeys had to remember the location (out of eight possibilities) of a briefly presented visual target for 2.3 s. One second after the target disappeared, a distractor was presented briefly in a different location. At the end of an additional 2.3 s, the monkeys reported the location of the remembered target using an eye movement (*Figure 1a*). We recorded single-unit activity from the LPFC and FEF of both monkeys while they performed the task. We only analyzed data collected for seven target locations for both animals, since one animal had difficulty making saccades to the lower-right location. *Figure 1b* shows the different electrode positions in the LPFC and FEF on an anatomical map. Additionally, FEF electrodes were differentiated from LPFC electrodes using microstimulation (see Materials and methods). We previously reported that the presentation of the distractor led to code-morphing in the LPFC (which was not observed in the FEF), such that a decoder trained in the delay period that preceded the distractor (Delay 1) could not be used to decode memory locations during the delay period that followed the distractor (Delay 2), and vice versa (*Parthasarathy et al., 2017*; *Figure 2a*). In other words, there were two stable population codes in the LPFC, one in Delay 1 and one in Delay 2, but they did not generalize to each other. In this paper, the presence of code-morphing in the LPFC motivated us to analyze the 226 single neurons recorded from the LPFC, which did not include those recorded from the FEF. Single neurons in the LPFC showed sustained selectivity to target locations during both delay periods, with some maintaining the same target tuning in both delays (*Figure 1c*, left), while some changed target tuning from Delay 1 to Delay 2 (*Figure 1c*, right). The latter category of neurons was characterized as non-linearly mixed selective neurons and was shown to drive code-morphing in the LPFC (*Parthasarathy et al., 2017*). On the population level, most of the cells with target selectivity in one delay also showed selectivity in the other delay (*Figure 1d*).

### Two minimally dependent subspaces coexisted within the LPFC

Two different and stable population activity patterns in the LPFC were observed in Delay 1 and Delay 2 (*Figure 2a*), which implied that a downstream region would need to use different decoders in the two periods to extract the working memory information (neural codes supporting the discrimination of different intended items), and would need to know which of them to use in the appropriate delay period. Alternatively, the difference observed between Delay 1 and Delay 2 activity could be explained by a superposition of different types of information in independent subspaces, such that each downstream region can use the same decoder to extract a specific type of information invariantly across time, even if the mixing of different types of information is dynamic across time. We have previously shown that a time-invariant (henceforth stable) working memory subspace can be identified in the LPFC (*Parthasarathy, 2019*). However, significant information about the target was present outside of this space (null space decoding performance of $35.7 \pm 1.7\%$ in Delay 1, and $31.6 \pm 1.5\%$ in Delay 2) (*Parthasarathy, 2019*) suggesting the existence of a non-trivial additional subspace that contains target information. The incorporation of the new information from the additional subspace into the neuronal population, alongside the existing information from the working memory subspace, would have then resulted in code morphing in the full space (illustrated in *Figure 2b*).

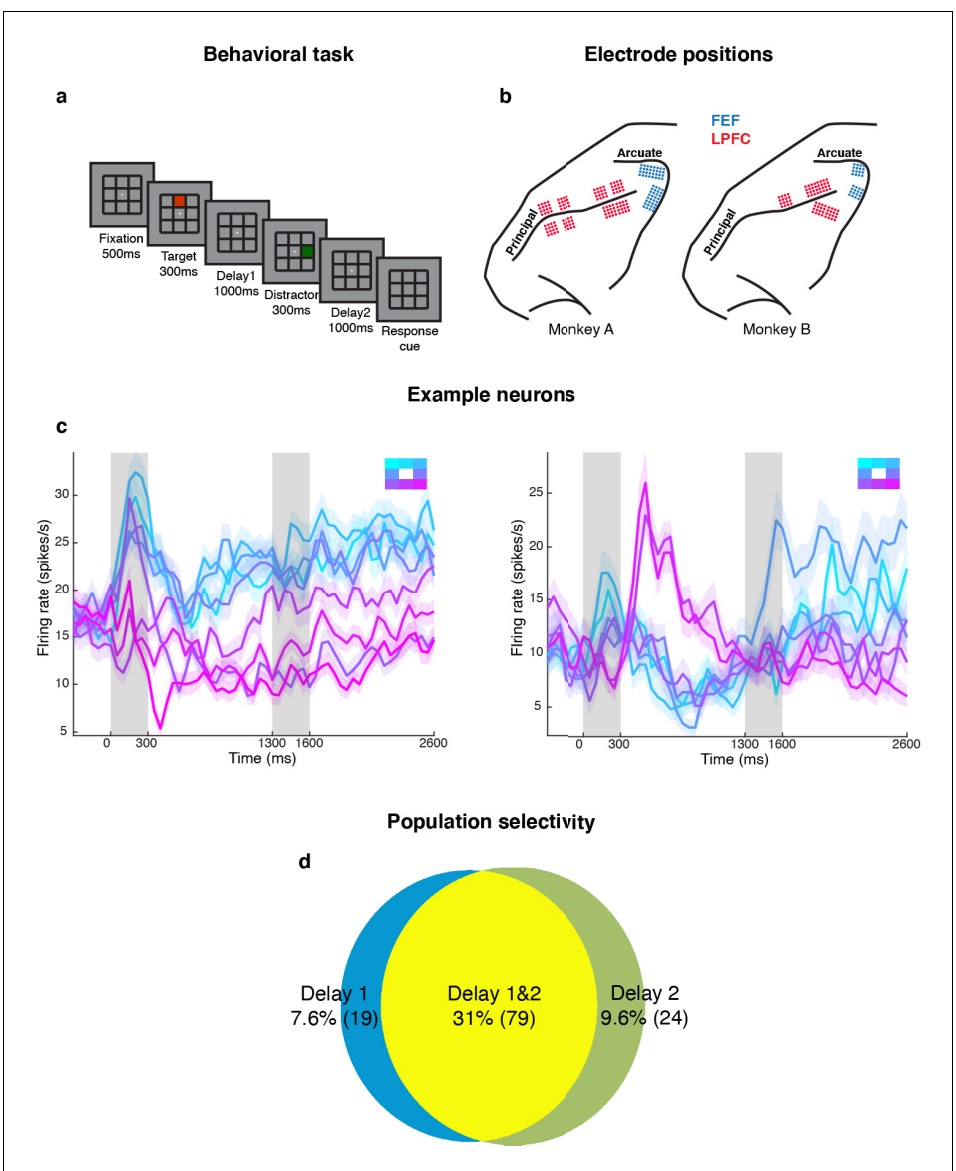

**Figure 1.** Experimental design and responses of example neurons. (a) Behavioral task: Each trial began when the animal fixated on a fixation spot at the center of the screen. The animal was required to maintain fixation throughout the trial until the fixation spot disappeared. A target (red square) was presented for 300 ms followed by a 1000 ms delay period (Delay 1). A distractor (green square) was then presented for 300 ms in a random location that was different from the target location and was followed by a second delay of 1000 ms (Delay 2). After Delay 2, the fixation spot disappeared, which was the Go cue for the animal to report, using an eye movement, the location of the target. (b) Implant locations of 16-channel and 32-channel electrode arrays (with electrode lengths ranging from 5.5 mm closer to the sulci, to 1 mm further from the sulci) in the LPFC (red dots) and the FEF (blue dots) in the two animals. Analyses were carried out only on LPFC data. (c) Peristimulus time histograms (PSTH) for two single neurons in the LPFC. Time 0 marks the onset of target presentation; responses to the different target locations are color-coded according to the legend shown in the top right; the colored regions surrounding each line indicates the standard error. (d) Venn diagram showing the number of LPFC neurons selective in Delay 1, in Delay 2, and their overlap. Target selectivity was tested using one-way ANOVA (p < 0.05) with spike counts averaged during 800–1300 ms for Delay 1 and 2100–2600 ms for Delay 2.

One possible source of the new information could be motor preparation activity. The stable population activity in Delay 2 suggested that the animals could have initiated preparatory activity right after distractor offset, as the distractor was the last signal that reliably predicted the Go cue. In order to assess this possibility, we looked for a subspace decomposition that could maximally differentiate

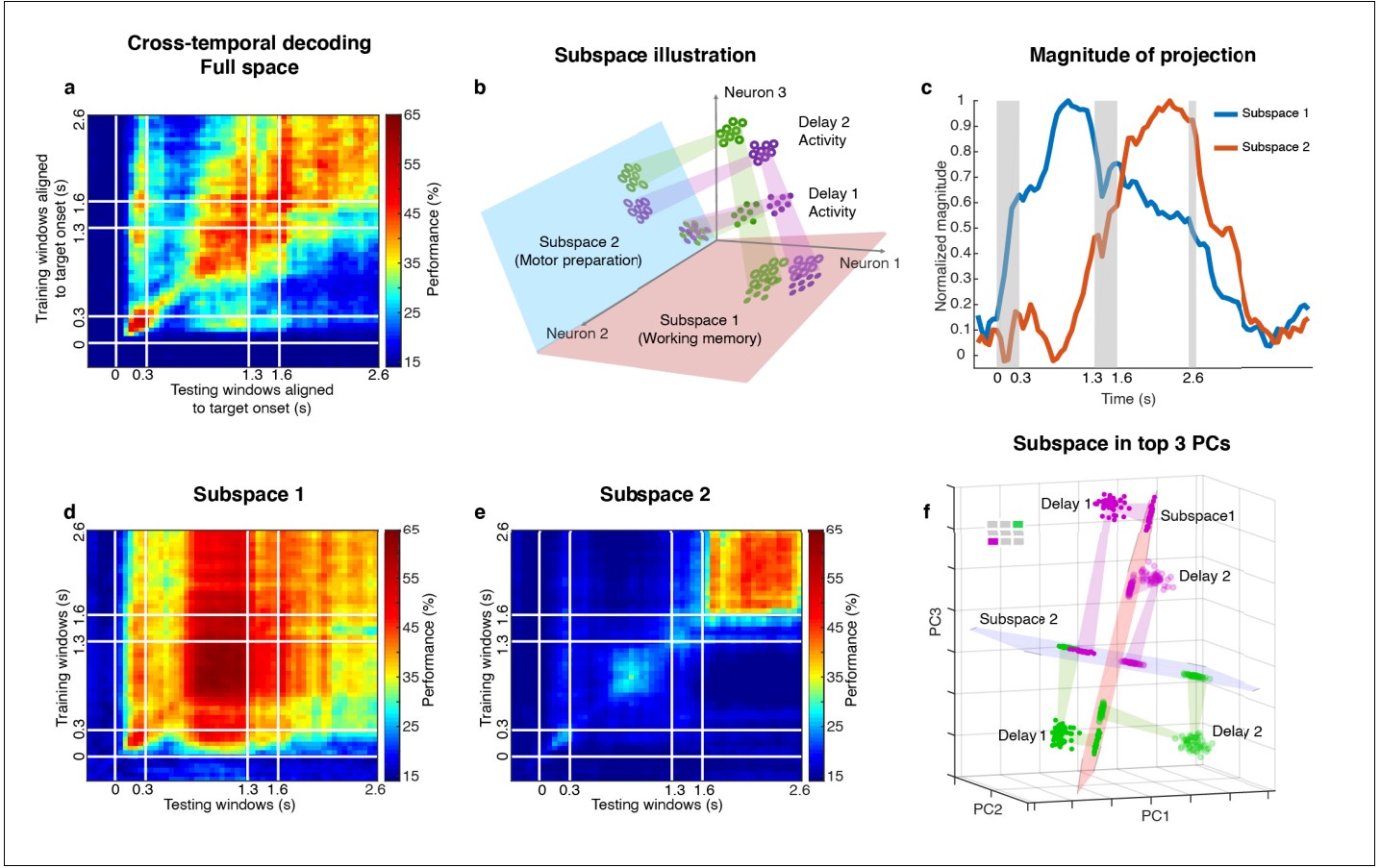

**Figure 2.** Code morphing, and two minimally dependent subspaces. (a) Heat map showing the cross-temporal population-decoding performance in the LPFC. White lines indicate target presentation (0–0.3 s), distractor presentation (1.3–1.6 s), and cue onset (2.6 s). (b) Schematic illustration of the projection of the full-space activity into *Subspace 1* and *Subspace 2*. Delay 1 activity (purple and green filled circles) projected into the *Subspace 1* would cluster according to target location (filled circles in the red subspace), and because this was a stable subspace, the Delay 2 activity for each target location (purple and green unfilled circles) would overlap with those for Delay 1 (open circles in the red subspace). In *Subspace 2,* Delay 1 activity would not cluster according to location (filled circles in the blue subspace), and the clustering by location would emerge only from the Delay 2 activity (open circles in the blue subspace) after the emergence of the new information. (c) We projected the trial-averaged full-space population activity for each time bin across the whole trial into *Subspace 1* and *Subspace 2* and calculated the magnitude of the projections. For each subspace, the magnitude was normalized to have a maximum value of 1. The projections into *Subspace 1* and *Subspace 2* exhibited different temporal profiles. (d) Cross-temporal decoding performance after projecting full-space activity into *Subspace 1*. (e) Cross-temporal decoding performance after projecting full-space activity into *Subspace 2*. (f) Projection of single-trial activity for two target locations (actual locations shown in the upper left corner) onto the first three principal components. Delay 1 is depicted as closed circles, and Delay 2 as open circles. Re-projections into the *Subspace 1* (red plane) and *Subspace 2* (blue plane) are shown and guided by projection cones (green and purple cones connecting the PCA projections into the subspace re-projections).

The online version of this article includes the following figure supplement(s) for figure 2:

**Figure supplement 1.** Unmixed population activity between Delay 1 and Delay 2.
**Figure supplement 2.** Single-session subspace identification.
**Figure supplement 3.** Effective dimension of full-space data in the subspaces.
**Figure supplement 4.** PCA projections in the first and second subspaces.
**Figure supplement 5.** Inter- and intra-cluster distance analysis.
**Figure supplement 6.** Mean population firing rate.
**Figure supplement 7.** Correlated and uncorrelated information.

the neural codes for working memory and motor preparation. Standard methods for such decompositions are regression (*Mante et al., 2013*; *Brody et al., 2003*) (resulting in a one-dimensional component for each task-dependent variable) and, more interestingly, Demixed Principal Component Analysis (dPCA) (*Brendel et al., 2011*; *Kobak et al., 2016*), which selectively isolates and constructs

a subspace for one task-dependent variable at a time by averaging out all the other task-dependent variables. However, these methods are not suitable for our data, because we always had the same target location label for working memory and motor preparation in each trial. As a result, for regression, the working memory and motor preparation variables will not have different coefficients; for dPCA, we cannot differentially represent the neural activity by averaging trials according to working memory or motor preparation locations to find different subspaces. Instead, we developed a novel method to identify the two subspaces even given that working memory and motor preparation always had the same target labels in each trial by regarding the trial-averaged and time-averaged Delay 1 and Delay 2 activity (each had a size 226 × 7, where 226 was the number of neurons, and seven was the number of target locations) as a mixture of working memory and motor preparation activity, and assumed that working memory and motor preparation activity themselves to be minimally dependent on each other. Our objective was then to find, through an optimization technique, the best unmixing coefficients to apply to Delay 1 and Delay 2 activity that could recover the working memory and motor preparation activity with the lowest mutual information possible between them (see Materials and methods). The original Delay 1 and Delay 2 activity exhibited 0.33 bits of mutual information. Using our method, we found two unmixed elements (representations of unmixed population activity that were each of size 226 × 7) from D1 and D2 activity with a minimum mutual information of 0.08 bits (*Figure 2—figure supplement 1*). The two elements we identified consisted of seven vectors in the 226-dimensional space, and according to the unmixing coefficients we identified, the magnitude of one element (*Element 1*) in Delay 2 was 65% of that in Delay 1, and the magnitude of the other element (*Element 2*) in Delay 1 was 12% of that in Delay 2. The orthonormal bases of the two elements defined two subspaces (*Subspace 1* and *Subspace 2*). The temporal dynamics of the full-space population activity projected into these subspaces showed that the magnitude of activity in *Subspace 1* increased early after target presentation and was maintained until the saccade cue, while the magnitude of activity in *Subspace 2* increased after distractor presentation and stayed relatively high even after the Go cue (*Figure 2c*, single-session results are shown in *Figure 2—figure supplement 2*). Next, we used the decoding performance of a linear decoder (LDA) as a proxy of target information and evaluated target information in each subspace. We trained an LDA decoder at each time point of the trial, and tested the decoder against all other time points across the trial to evaluate the temporal generalization of the population activity (cross-

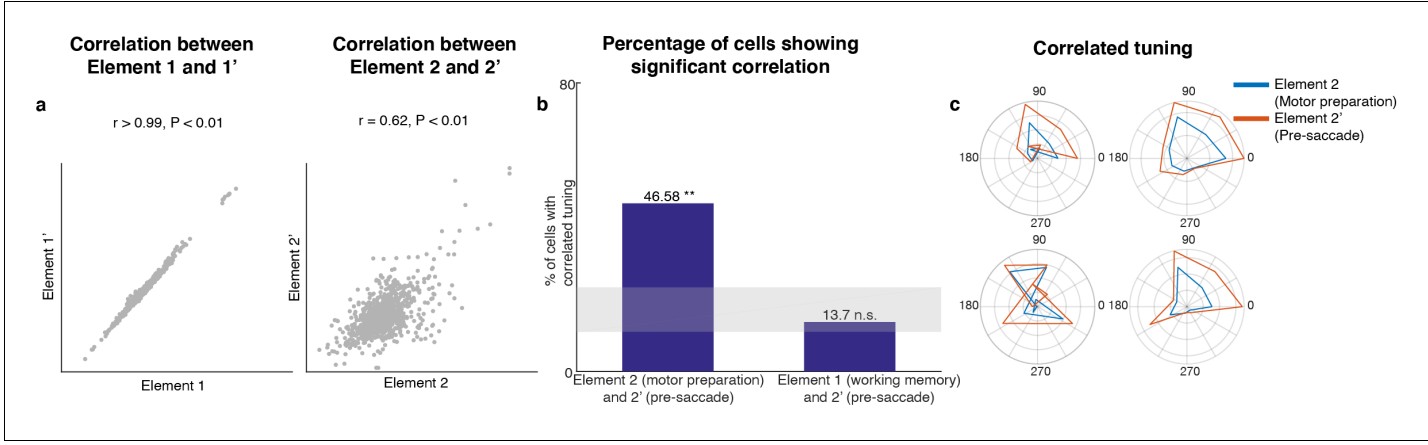

**Figure 3.** Preparatory and pre-saccadic activity. (a) Correlation between unmixed elements found from Delay 1/Delay 2 activity (*Elements 1* and *2*) and the unmixed elements from Delay 1/pre saccadic activity (*Elements 1'* and *2'*). The matrices were flattened into 1-d for the correlation analysis. *Elements 1* and *1'* were almost identical (r > 0.99, p < 0.01, left), while *Elements 2* and *2'* were highly correlated (r = 0.62, p < 0.01, right). (b) Left, the percentage of cells that exhibited significant correlation between *Elements 2* and *2'*. Right, the percentage of cells that exhibited significant correlation between *Elements 1* and *2'*. The shaded area shows the 5th and 95th percentiles of the chance percentage obtained by shuffling the tuning across cells. (c) Response tuning of four representative cells that showed significant correlation between their activity in *Elements 2* and *2'*.

The online version of this article includes the following figure supplement(s) for figure 3:

**Figure supplement 1.** Principal angles between subspaces.

**Figure supplement 2.** Decorrelated population activity between Delay 1 and the pre-saccadic period.

**Figure supplement 3.** Cross-temporal decoding for distractor locations.

temporal decoding, see Materials and methods). Cross-temporal decoding after projecting the full-space neural activity into *Subspace 1* showed that information emerged right after target presentation, and although the information was higher in Delay 1 (60.5 ± 1.3%), it was present throughout the whole trial, even during the distractor period (*Figure 2d*). This was qualitatively consistent with our hypothesis, aside from the decrease in information in Delay 2 (39.9 ± 1.1%). Cross-temporal decoding of full-space neural activity projected into *Subspace 2* showed that information emerged after distractor presentation (42.6 ± 1.1%), and was stable throughout Delay 2 (*Figure 2e*). In Delay 1, *Subspace 1* and *Subspace 2* accounted for 14.6% and 10.3% of the variance in the full space; in Delay 2, *Subspace 1* and *Subspace 2* accounted for 5.8% and 8.1% of the variance in the full space. Full-space data in the two subspaces had an effective dimensionality of 6 dimensions each – after projecting single-trial full-space data into the subspaces, we performed a PCA on the projected data, and the first six out of the seven principal components cumulatively accounted for more than 95% of the variance within each subspace (*Figure 2—figure supplement 3*). This indicated that the true dimensionality of the neural code could be smaller than the number of discrete target locations imposed by the experiment. In addition, as the number of discrete target locations increases in the experiment (for example, 24 target locations), we expect the effective dimensionality of data in the subspaces will asymptote to the true dimensionality of the neural codes supporting the cognitive processes.

*Figure 2f* shows single-trial projections of two different target locations (purple and green locations shown in the top-left corner) onto the top three principal components (PCs). These projections were then re-projected into *Subspace 1* (red plane) and *Subspace 2* (blue plane). The low-dimensional visualizations are merely used to provide intuitions underlying the cross-temporal decoding results, which were all obtained using high-dimensional data (see Materials and methods). Consistent with our hypothesis, Delay 1 and Delay 2 projections into *Subspace 1* clustered according to target location, although they overlapped less than we expected (we will revisit this deviation from our expectation later on). However, the separation between the projected points was small enough that target location information could be decoded in both delays, regardless of whether the classifier was trained using Delay 1 or Delay 2 activity (*Figure 2d*).

On the other hand, projections into *Subspace 2* behaved differently, such that Delay 1 projections for multiple target locations overlapped, whereas Delay 2 projections remained separated. This explained why in *Subspace 2*, target location information could not be decoded in Delay 1, but could be decoded in Delay 2 (*Figure 2e*). Projections into *Subspace 1* and *Subspace 2* for all target locations confirmed that these observations generalized to the rest of the locations (*Figure 2—figure supplement 4*, which also illustrates the reason for the difference in performance in the two off-diagonal quadrants in *Figure 2d*).

## The two minimally dependent subspaces corresponded to working memory and motor preparation

Since *Subspace 1* contained target information throughout the trial, and working memory of the target location was presumably required throughout the trial, we hypothesized that *Subspace 1* corresponded to a working memory subspace. We previously showed that the LPFC contained a working memory subspace that encoded stable working memory information (*Parthasarathy, 2019*). In order to assess whether *Subspace 1* corresponded to the working memory subspace previously described (*Parthasarathy, 2019*), we calculated the principal angles between these subspaces, as a measure of similarity (see Materials and methods). We found that *Subspace 1* was significantly closer than chance to the working memory subspace, while *Subspace 2* was not (*Figure 3—figure supplement 1*). This result supported the interpretation that *Subspace 1* corresponded to a working memory subspace. Thus, henceforth, we will refer to *Subspace 1* as the 'working memory subspace' and *Element 1* as the unmixed working memory element.

Since *Subspace 2* contained target information only after the distractor disappeared, and motor preparation presumably began after the last sensory cue that reliably predicted the timing of the Go cue (i.e. the offset of the distractor), we hypothesized that *Subspace 2* corresponded to a motor preparation subspace. Activity between the Go cue and the saccade onset contained information about saccade execution (45% of LPFC neurons we recorded were selective in the period between the Go cue and saccade onset, assessed using a one-way ANOVA, $p < 0.05$). In order to test whether *Subspace 2* corresponded to a motor preparation subspace, we compared the original

unmixed elements using Delay 1 and Delay 2 activity with a new pair of unmixed elements using Delay 1 and pre-saccadic period activity (150 ms to 0 ms prior to saccade). If *Subspace 2* corresponded to a motor preparation subspace, we should observe similarities between the second element in both pairs of unmixed elements, that is the unmixed motor preparation activity and the unmixed pre-saccade activity (see Materials and methods). In the new pair of unmixed elements, we obtained two elements with relative vector magnitudes similar to those found in the first pair of unmixed elements (for *Element 1'*, the vector magnitude in the pre-saccade period was 70% of that in Delay 1, while for *Element 2'*, the vector magnitude in Delay 1 was 0% of that in the pre-saccade period, *Figure 3—figure supplement 2*). We found that *Element 1* and *Element 1'* were significantly correlated (*Figure 3a* left, Pearson correlation $r > 0.99$, $p < 0.01$). Importantly, *Element 2* and *Element 2'* were also significantly correlated (*Figure 3a* right, Pearson correlation $r = 0.62$, $p < 0.01$). This result supported our hypothesis that *Subspace 2* corresponded to a motor preparation subspace.

In an additional test of the hypothesis that *Subspace 2* corresponded to a motor preparation subspace, we examined the relationship between the unmixed motor preparation activity and the unmixed pre-saccade activity at the level of single cells. First, we identified cells with spatial tuning in both Delay 2 and the pre-saccade period (73 cells, two one-way ANOVAs, both $p < 0.05$). Then, for each cell, we measured the correlation between the unmixed motor preparation activity and the unmixed pre-saccade activity across different target locations. We found that 47% of these neurons showed significant correlation (Pearson correlation, $p < 0.05$), which exceeded the number expected by chance (*Figure 3b*, left bar, $p < 0.001$, $g = 10.82$). As a control, we carried out the same analysis between the unmixed working memory activity and the unmixed pre-saccade activity, and found no evidence of a higher number of correlated cells than expected by chance (*Figure 3b*, right bar, $p > 0.19$, $g = 1.51$). Examples of neurons with significant correlation are shown in *Figure 3c*. This result provided additional support to our hypothesis that *Subspace 2* corresponded to a motor preparation subspace. Thus, henceforth, we will refer to *Subspace 2* as the 'motor preparation subspace'. Alongside the working memory and motor preparation activity for target locations, there could also be activity representing distractor locations in Delay 2. By grouping trials according to distractor labels, we indeed found significant distractor information in the full space (*Figure 3—figure supplement 3*). However, the distractor activity in Delay 2 was not related to the *Element 2* or the motor preparation subspace we identified, because the distractor activity and the motor preparation activity were obtained from data grouped by different trial labels (target and distractor labels were uncorrelated). Very little distractor information (17.9 ± 0.7%) was successfully decoded in the motor preparation subspace (*Figure 3—figure supplement 3*).

## Activity of neurons with mixed working memory and motor preparation selectivity formed the two subspaces

The existence of two minimally dependent subspaces could be mediated by one of two possible mechanisms: (1) two distinct subpopulations of neurons with exclusive working memory or motor preparation selectivity within the LPFC, or (2) the same population of LPFC neurons with mixed selectivity to both working memory and motor preparation. In order to distinguish between these two possible mechanisms, we projected the unit vector representing each neuron in the full space into the working memory and motor preparation subspaces, and quantified the magnitude of the two projections for each neuron (i.e. loading weight, *Figure 4a*). A clustering of the loading weights along the x- and y-axes would support the first mechanism, and if not this case, a non-significant or positive correlation would support the second mechanism. In order to test if the points clustered along the x- and y-axes, we computed the ratio of points near the x- or y-axes (above 67.5° line or below 22.5° line) for 1,000 bootstraps, with a random 10% data exclusion in each bootstrap. We found that there were significantly more points away from the x- or y-axes (88% to 90%, corresponding to the 5th and 95th percentile of 1,000 bootstraps), which rejected the first mechanism. In addition, to our surprise, we found a significant positive correlation between the loading weights in each subspace ($r = 0.68$, which was significantly higher than the 95th percentile of 1,000 shuffles, where in each shuffle we randomly permuted the population's loading weights for both subspaces), which not only supported the second mechanism, but further suggested that neurons with stronger contribution to the working memory subspace would also have a stronger contribution to the motor preparation subspace. As a measure of the relative contribution to each subspace, we calculated the ratio

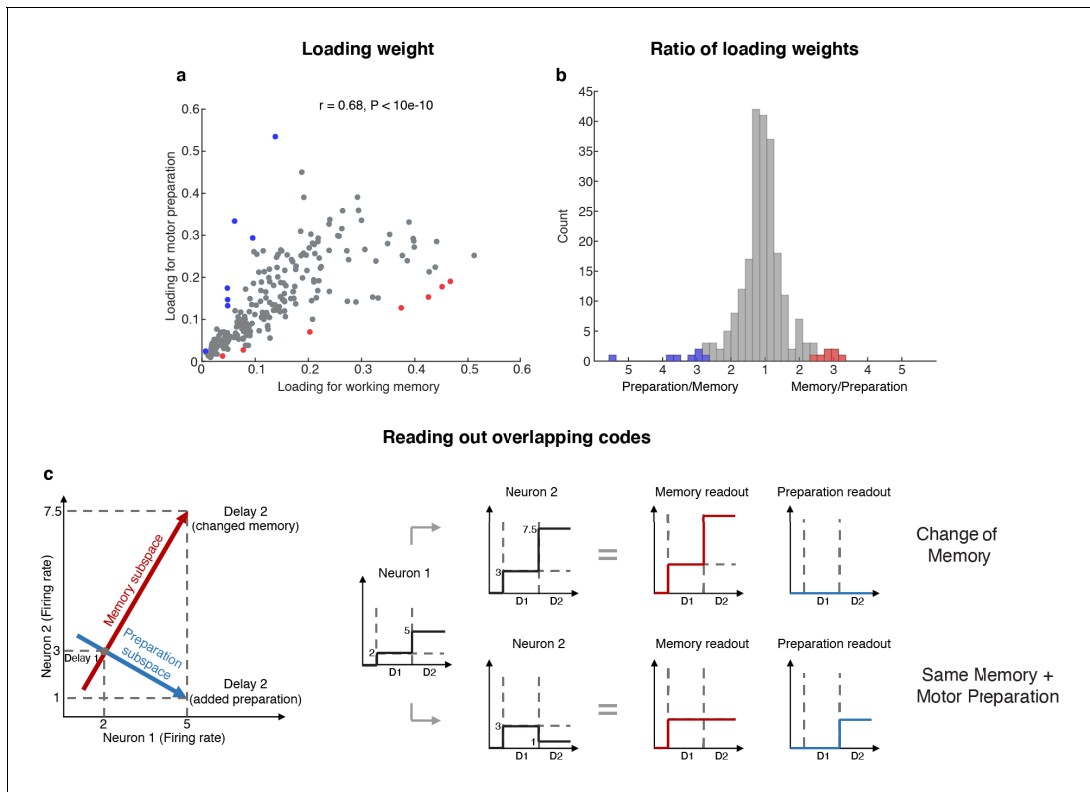

**Figure 4.** Loading weights for individual neurons. (a) The loading weight of each neuron in the working memory subspace and the motor preparation subspace. (b) Histogram of the ratio between the loading weights for each cell. For cells with larger loading for the working memory subspace, the values are plotted to the right of the plot, while for cells with larger loading for the motor preparation subspace, the values are plotted to the left of the plot. Red dots (in a) and bars (in b) represent cells with 'exclusive' loading for the working memory subspace, while blue dots (in a) and bars (in b) represent cells with 'exclusive' loading for the motor preparation subspace. These cells were identified as those with ratios that exceeded two standard deviations from the mean. (c) Illustration of how overlapping codes can be read out. Different loading weights of the two subspaces (expressed as connection weights between the readout neurons and Neurons 1 and 2) unambiguously read out working memory or motor preparation information from the conjunctive population code formed by both *Neuron 1* and *Neuron 2*, whereas it would have been ambiguous to look only at *Neuron 1*'s change of firing rate in Delay 2.

The online version of this article includes the following figure supplement(s) for figure 4:

**Figure supplement 1.** Cross-temporal decoding on the population with mixed selectivity and populations with exclusive selectivity.
**Figure supplement 2.** Bump attractor models with and without normalization.
**Figure supplement 3.** Neuronal selectivity.
**Figure supplement 4.** Linear subspace model.

between the loading weights for each cell, and analyzed their distribution (*Figure 4b*). We found that only 14 (6%) of the neurons had 'exclusive' loading for the working memory (red) or motor preparation (blue) subspaces. However, these cells were not necessary to identify the subspaces (*Figure 4—figure supplement 1*).

In order to understand how a single population of neurons with mixed selectivity could have contributed minimally dependent information to the two subspaces, we created a simple illustration (*Figure 4c*). Working memory and motor preparation information were read out by separate readout neurons with different connection weights to Neurons 1 and 2 that reflected the loading weights of each subspace. In isolation, the activity of *Neuron 1* would be ambiguous for both readout neurons, as an increase of activity in Delay 2 could be interpreted as a new memory at a different spatial location, or as the same memory as in Delay 1, but with superimposed motor preparation activity. In order to disambiguate the meaning of a change in the activity of one neuron, it would be necessary to interpret that change in the context of changes in the activity of the rest of the neuronal population (i.e. in this example, *Neuron 2*). In the illustration, a superimposed increase of activity in Neurons 1 and 2 signals a change in memory (i.e. only the readout activity in the working memory subspace

changed), whereas the same increase in *Neuron 1*, but with a superimposed decrease of activity in *Neuron 2*, signals that the memory has not changed, but that a motor preparation plan has emerged in Delay 2 (i.e. only the readout activity in the motor preparation subspace changed). This concept can be extended to the 212 neurons with mixed selectivity to understand how the coordinated activity between those neurons can contribute minimally dependent information to the working memory and motor preparation subspaces through different loading weights that we found in the LPFC (low-dimensional visualizations of neural data provided in *Figure 2f* and *Figure 2—figure supplement 4*).

## Information in one subspace led to a small amount of interference in information in the other subspace

Since one population of neurons with mixed selectivity contributed to both the working memory and motor preparation subspaces, it was possible that information in one of the subspaces interfered with information in the other, and vice versa. We checked if the two subspaces were orthogonal to each other by comparing their principal angles with those between two random subspaces of the same dimension (*Figure 5a*). All the principal angles between the working memory and motor preparation subspaces were significantly smaller than chance, indicating non-orthogonality between the two subspaces and the likelihood of interference of information between them. One way to assess the interference between the *information* in both subspaces is to ask whether the emergence of motor preparation activity in Delay 2 (added on top of working memory activity) changes the amount of readable information in the working memory subspace, compared to the situation where there is only working memory activity in Delay 2. In order to quantify this interference of information, we compared the decoding performance of an LDA classifier trained and tested on the unmixed single-trial working memory activity (see Materials and methods) projected into the working memory subspace (*Figure 5a*: $proj_{MSub}(M)$), with decoding performance of a classifier trained and tested on single-trial working memory activity plus motor preparation activity (i.e. Delay 2 activity) projected into the working memory subspace (*Figure 5a*: $proj_{MSub}(M+P)$). A similar analysis was carried out in the motor preparation subspace (*Figure 5b*: $proj_{PSub}(P)$ and $proj_{PSub}(M+P)$). We found no evidence of a drop in performance between $proj_{MSub}(M)$ and $proj_{MSub}(M+P)$ ($p > 0.73$, $g = 0.61$), and between $proj_{PSub}(P)$ and $proj_{PSub}(M+P)$ ($p > 0.22$, $g = 2.63$), suggesting a lack of interference between these subspaces.

As we used LDA's decoding performance as a proxy of target information, the lack of interference between two non-orthogonal subspaces indicated that the shift of clusters in the state space caused by superimposed activity were not large enough to cross the classification boundaries, and thus did not affect the classification performance. So we performed a more sensitive state-space analysis on Delay 2 activity to assess whether the working memory and motor preparation subspaces interfered with each other. We quantified interference by projecting the unmixed single-trial activity into the two subspaces, and calculated the average distance between clusters of points corresponding to different target locations (inter-cluster distance). The inter-cluster distance was then normalized by the average intra-cluster distance for all clusters, which was a measure of trial-by-trial variability in the population activity (see Materials and methods). This inter-to-intra cluster distance ratio was compared between projections of unmixed single-trial working memory activity into the working memory subspace (*Figure 5d*: $proj_{MSub}(M)$), and projections of single-trial working memory plus motor preparation activity (i.e. Delay 2 activity) into the working memory subspace (*Figure 5d*: $proj_{MSub}(M+P)$). A similar analysis was carried out in the motor preparation subspace (*Figure 5e*). We found a small decrease (7.1%) of the inter-to-intra cluster distance ratio when both working memory and motor preparation activity were projected into the subspaces, suggesting the existence of a small, but significant interference between both subspaces ($p < 0.001$, $g = 4.81$ between $proj_{MSub}(M)$ and $proj_{MSub}(M+P)$, $p < 0.001$, $g = 6.63$ between $proj_{PSub}(P)$ and $proj_{PSub}(M+P)$).

## Less information was found in error trials in both subspaces

In a subset of trials, the animals maintained fixation until the Go cue, but failed to report the correct target location with a saccade. These failures could be due to the animals reporting other locations, including the location of the distractor, or simply saccading to a completely different location, such as the edge of the monitor. Classifiers trained on unmixed single-trial working memory activity of

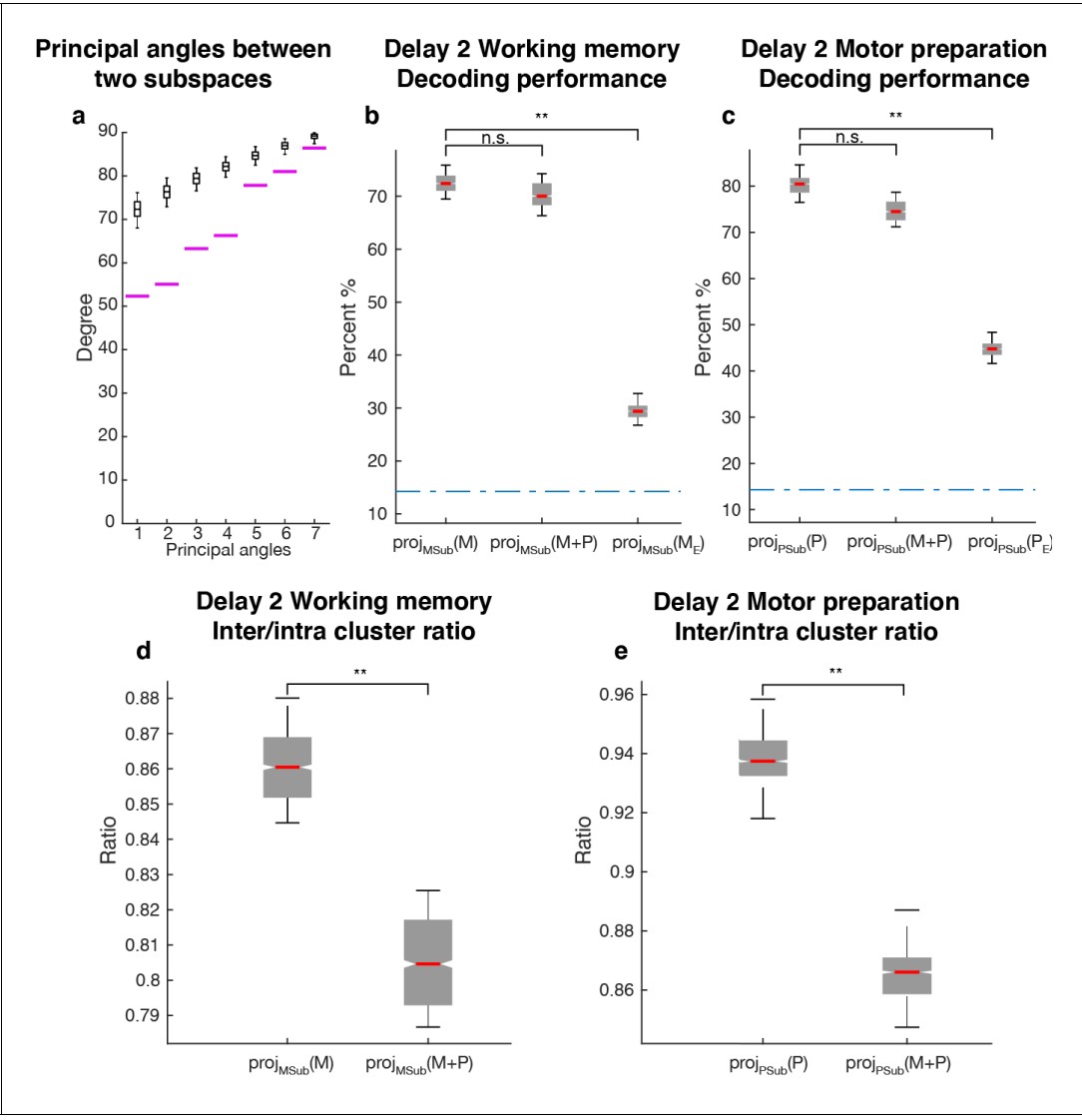

**Figure 5.** Comparisons between working memory and motor preparation subspaces. (a) The principal angles between the working memory subspace and the motor preparation subspace are shown in the magenta lines in ascending order. The black boxplots illustrate the distribution of principal angles between two random subspaces with the same dimensions as the working memory and motor preparation subspaces. The borders of the box represented the 25[th] and the 95[th] percentile of the distribution, while the whiskers represent the 5[th] and the 95[th] percentiles. (b) Decoding performance in the working memory subspace. $M$ stands for unmixed single-trial working memory activity; $proj_{MSub}(M)$, decoding of the unmixed single-trial working memory activity projected into the working memory subspace; $proj_{MSub}(M+P)$, decoding of the single-trial Delay 2 activity projected into the working memory subspace; $proj_{MSub}(M_E)$, decoding of the unmixed single-trial working memory activity in error trials projected into the working memory subspace using a classifier built on unmixed single-trial working memory activity in correct trials projected into the working memory subspace; (c) Decoding performance in the motor preparation subspace. $P$ stands for unmixed single-trial motor preparation activity. Same conventions as in a, but for unmixed single-trial motor preparation activity and the motor preparation subspace. We verified that the drop in performance in error trials was specific to the two subspaces, and not due to a non-specific increase in noise in the population (see Materials and methods). (d) Inter-to-Intra cluster ratio of unmixed single-trial working memory activity projected into the working memory subspace ($proj_{MSub}(M)$), and of single-trial full-space activity projected into the working memory subspace ($proj_{MSub}(M+P)$). (e) Same conventions as in d, but for unmixed single-trial motor preparation activity and motor preparation subspace.

The online version of this article includes the following figure supplement(s) for figure 5:

**Figure supplement 1.** Gram-Schmidt orthogonal decomposition.

**Figure supplement 2.** | Analytical memory subspace and non-memory subspace.

**Figure supplement 3.** | Amount of interference in different methods.

correct trials projected into the working memory subspace, proj$_{MSub}$(M), were tested on unmixed single-trial working memory activity of error trials (see Materials and methods), which was also projected into the working memory subspace, proj$_{MSub}$(M$_E$). Decoding performance was significantly reduced in error trials (*Figure 5a,*) compared to correct trials (*Figure 5a*, p < 0.001, *g* = 13.9 between proj$_{MSub}$(M) and proj$_{MSub}$(M$_E$)), suggesting that failures in memory encoding occurred during error trials. A similar analysis on the motor preparation subspace yielded equivalent results (*Figure 5b*, p < 0.001, *g* = 13.8 between proj$_{PSub}$(P) and proj$_{PSub}$(P$_E$)), which were consistent with the fact that in error trials, saccades were made to different locations than in correct trials. These results suggested that the subspaces we found could have been used by the animals to perform the task.

## Artificial neural networks with divisive normalization recapitulated the properties of LPFC activity

An unexpected observation in our results was that decoding performance in the working memory subspace decreased in Delay 2 compared to Delay 1 (*Figure 2d*). This decrease coincided with an increase of decoding performance in the motor preparation subspace (*Figure 2e*). The reduction of working memory decoding performance was not expected in the schematic diagram of the subspace dynamics (*Figure 2b*), but was captured by the state space visualization of real neural data (*Figure 2f*, inter-cluster distance in *Subspace 1* reduced in Delay 2). A more rigorous state space analysis revealed that the decrease of working memory decoding performance was due to a decrease in the inter-to-intra cluster distance ratio of working memory activity in Delay 2, and the increase of motor preparation decoding performance was due to an increase in the inter-to-intra cluster distance ratio of motor preparation activity in Delay 2 (*Figure 2—figure supplement 5*). In addition, we noticed that the mean population firing rate did not change between Delay 1, Delay 2 and pre-target fixation periods (*Figure 2—figure supplement 6*). This observation was consistent with a population normalization mechanism to maintain the mean population firing rate at a constant level in the LPFC (*Ruff and Cohen, 2017*; *Duong et al., 2019*). In order to assess whether a normalization mechanism was responsible for the decrease of working memory information in Delay 2, we built artificial neural network models with and without population normalization, and compared their behavior with the LPFC data.

Bump attractor models have been shown to replicate several properties of LPFC activity, including code-morphing in the full space, the existence of a stable subspace with stable working memory information, and non-linear mixed selectivity of individual neurons (*Parthasarathy, 2019*; *Compte et al., 2000*; *Wimmer et al., 2014*). Here, we created a model that incorporated subsets of neurons that represented information in the working memory and motor preparation subspaces (*Figure 4—figure supplement 2*), and looked at the effect of adding divisive normalization to keep the mean population firing rate constant. We constrained the attractor model to utilize neurons with mixed selectivity by matching the selectivity properties to those found in the LPFC data (*Figure 4—figure supplement 3*). We also used the same unmixing method to identify working memory and motor preparation subspaces from the activity in the attractor model. We found that only the attractor model with divisive normalization qualitatively replicated all the features of the neural data (*Figure 4—figure supplement 2*). We also explored an alternative model that supported the implementation of subspaces – a linear subspace model (*Murray et al., 2017*). Similarly, only the linear subspace model with divisive normalization replicated all the main findings in the neural data (*Figure 4—figure supplement 4*). The results of the model simulations supported the idea that divisive normalization was needed to faithfully replicate the properties of neural data, and suggested that such a mechanism constituted part of the function of the LPFC. The similarity between the results obtained from two different categories of models – the bump attractor model and the linear subspace model, also indicated the conceptual convergence between the two models in replicating the sustained-activity aspect of the LPFC activity.

## Discussion

Here, we demonstrate that two minimally dependent subspaces coexist within the LPFC population. These subspaces contain largely independent information about target location, and appear to encode working memory and motor preparation information. We show that there is a small, but

significant interference of information when both subspaces encode information simultaneously, and during error trials information in the working memory subspace is reduced. Assessment of activity properties of individual neurons revealed that a single population of neurons with mixed selectivity generates both subspaces. Finally, we show that a bump attractor neural network model with divisive normalization can capture all these properties described. Overall, our results show that working memory and motor preparation subspaces coexist in a single neural network within the LPFC.

Our results provided the first evidence suggesting that information about two separate cognitive processes can be simultaneously encoded in subspaces within the same brain region. The majority of the literature on information subspaces in the brain has reported a single subspace (*Parthasarathy, 2019*; *Druckmann and Chklovskii, 2012*; *Inagaki et al., 2019*; *Svoboda and Li, 2018*), multiple subspaces in different regions (*Semedo et al., 2019*), or multiple subspaces in the same region where information transited from one subspace to another but did not coexist simultaneously in different subspaces (*Kaufman et al., 2014*; *Elsayed et al., 2016*; *Yoo and Hayden, 2020*; *Kimmel et al., 2020*). *Mante et al., 2013*. showed coexistence of information in three subspaces in the prefrontal cortex, but two of the subspaces encoded color and motion (stimulus subspace), and only one subspace encoded action choice (cognitive subspace). *Minxha et al., 2020*. reported the existence of two cognitive subspaces for 'memory' and 'categorization' tasks, but the two subspaces were employed asynchronously in different types of trial blocks. Thus, the identification of two coexisting cognitive subspaces (working memory and motor preparation) in our work could provide new insights into the brain's mechanisms underlying our cognitive flexibility.

It is important to minimize interference between different types of information. For example, a visual area may read out working memory information (*Yeterian et al., 2012*; *Merrikhi et al., 2017*), while a premotor region may read out motor preparation information from the LPFC (*Yeterian et al., 2012*; *Churchland et al., 2012*; *Schall and Hanes, 1993*). If large interference existed between subspaces, the computations of downstream regions would be compromised. We found a small, but significant interference between the subspaces, such that some working memory information was reflected in the motor preparation subspace (and vice versa). It is not surprising that there is some degree of interference, since the method we used to decompose the signals did not impose a constraint to ensure maximal orthogonality between subspaces, and while the mutual information was low, it was not zero. In order to assess whether imposing orthogonality between subspaces was feasible, we fixed Delay 1 activity as the first activity subspace and rotated each column vector in Delay 2 activity matrix to be orthogonal to Delay 1 activity (Gram-Schmidt algorithm) to obtain the second activity matrix, such that the two activity matrices would be orthogonal to each other. To our surprise, the two simple orthogonal subspaces were highly similar to our working memory and motor preparation subspaces (*Figure 5—figure supplement 1*). The reason for this similarity could be that the unmixing method showed that there was very little motor preparation activity in Delay 1, so the assumption that Delay 1 activity exclusively contains working memory information would largely align with the unmixing results. However, the unmixing method was in principle a more flexible approach to identify the subspaces, and also provided a less constrained viewpoint to interpret the neural data (details discussed in *Figure 5—figure supplement 1*). We also considered alternative methods to identify the subspaces, but these produced subspaces with larger interference (*Figure 5—figure supplements 2* and *3*). The interference we found suggests that under conditions that stress the working memory and motor preparation systems (such as a task that requires the concurrent memorization of four targets, and preparation of four movements) a predictable bias should be observable for both the recalled target locations and eventual movements. This prediction remains to be tested.

We also found an indirect way in which information in subspaces interfere with each other: divisive normalization of population activity. This led to a decrease of working memory information in Delay 2 once motor preparation information emerged. Divisive normalization, which has been described before in the LPFC (*Ruff and Cohen, 2017*; *Duong et al., 2019*), could be useful as an energy saving mechanism, since it maintains the population activity at a low level when new information is added. A bump attractor model and a linear subspace model with divisive normalization allowed us to replicate the properties of LPFC activity. However, the models only provide high-level support for this mechanism, and a mechanistic implementation is still needed.

In this work, we derived two subspaces, and analyzed the benefits of decoding from those subspaces, from data in which the memory location and the motor preparation location were identical.

However, there are situations where the LPFC is required to store multiple pieces of information that are uncorrelated, for example if the animal has to remember the location and color of a target to perform a task, where these are uncorrelated (*Warden and Miller, 2010*; *Cavanagh et al., 2018*). We have verified that our approach can be extended to identify the relevant subspaces in tasks with uncorrelated information as well (see Materials and methods). We show that in tasks with uncorrelated information, decoding in the full space could result in higher interference as compared to tasks with correlated information (*Figure 2—figure supplement 7a,b*). However, the use of subspaces could reduce interference in both cases (*Figure 2—figure supplement 7c,d*), suggesting a possible advantage of encoding information in minimally dependent subspaces in a broad range of cognitive tasks that we typically associate with the LPFC.

Our results support a framework in which low-dimensional subspaces could be a general property of cortical networks (*Remington et al., 2018*; *Ruff and Cohen, 2019*). Under this framework, downstream regions could extract specific information from these subspaces (*Semedo et al., 2019*). This could provide a mechanism for selective routing of information between regions (*Yang et al., 2016*), which could in turn be a building block of our cognitive flexibility capacity. The dimensionality of a network's full state space constrains the number and dimensionality of the different information subspaces that could coexist within the network. The LPFC is a densely connected brain hub, anatomically connected to more than 80 regions, compared to the roughly 30 connected to the primary visual cortex (*Markov et al., 2014*). Given the variety of inputs to the LPFC, it is not surprising that a large number of its neurons show mixed selective activity (*Rigotti et al., 2013*; *Parthasarathy et al., 2017*), which in turn endow the LPFC with high dimensionality (*Rigotti et al., 2013*). This property would allow a higher number of subspaces to coexist within the network. The number of items that can be simultaneously maintained in working memory is limited, which may be the result of encoding constraints within the working memory subspace. However, the types of information that can be encoded in working memory seem limitless. These include memories of external events (such as the visual stimuli in the current study), as well as memories of internally-generated events (such as the task rules stored in long-term memory in the current study, which previous studies have shown to be reflected in LPFC activity; *Wallis et al., 2001*). It is possible that underlying this ability to encode such diverse types of information is the extremely large number of possible information subspaces that could coexist within regions of the LPFC.

# Materials and methods

## Subjects and surgical procedures

We used two male adult macaques (*Macaca fascicularis*), Animal A (age 4) and Animal B (age 6), in the experiments. All animal procedures were approved by, and conducted in compliance with the standards of the Agri-Food and Veterinary Authority of Singapore and the Singapore Health Services Institutional Animal Care and Use Committee (SingHealth IACUC #2012/SHS/757). The procedures also conformed to the recommendations described in Guidelines for the Care and Use of Mammals in Neuroscience and Behavioral Research (*Van Sluyters and Obernier, 2003*). Each animal was implanted first with a titanium head-post (Crist Instruments, MD) before arrays of intracortical microelectrodes (MicroProbes, MD) were implanted in multiple regions of the left frontal cortex. In Animal A, we implanted six arrays of 16 electrodes and one array of 32 electrodes in the LPFC, and two arrays of 32 electrodes in the FEF, for a total of 192 electrodes. In Animal B, we implanted one array of 16 electrodes and two arrays of 32 electrodes in the LPFC, and two arrays of 16 electrodes in the FEF, for a total of 112 electrodes. The arrays consisted of platinum-iridium wires with either 200 or 400 μm separation, 1–5.5 mm of length, 0.5 MΩ of impedance, and arranged in $4 \times 4$ or $8 \times 4$ grids. Surgical procedures followed the following steps. 24 hr prior to the surgery, the animals received a dose of Dexamethasone to control inflammation during and after the surgery. They also received antibiotics (amoxicillin 7–15 mg/kg and Enrofloxacin 5 mg/kg) for 8 days, starting 24 hr before the surgery. During surgery, the scalp was incised, and the muscles retracted to expose the skull. A craniotomy was performed (~2×2 cm). The dura mater was cut and removed from the craniotomy site. Arrays of electrodes were slowly lowered into the brain using a stereotaxic manipulator. Once all the arrays were secured in place, the arrays' connectors were secured on top of the skull using bone cement. A head-holder was also secured using bone cement. The piece of bone removed during the

craniotomy was repositioned to its original location and secured in place using metal plates. The skin was sutured on top of the craniotomy site, and stitched in place, avoiding any tension to ensure good healing of the wound. All surgeries were conducted using aseptic techniques under general anesthesia (isoflurane 1–1.5% for maintenance). The depth of anesthesia was assessed by monitoring the heart rate and movement of the animal, and the level of anesthesia was adjusted as necessary. Analgesics were provided during post-surgical recovery, including a Fentanyl patch (12.5 mg/2.5 kg 24 hr prior to surgery, and removed 48 hr after surgery), and Meloxicam (0.2–0.3 mg/kg after the removal of the Fentanyl patch). Animals were not euthanized at the end of the study.

## Recording techniques

Neural signals were initially acquired using a 128-channel and a 256-channel Plexon OmniPlex system (Plexon Inc, TX) with a sampling rate of 40 kHz. The wide-band signals were band-pass filtered between 250 and 10,000 Hz. Following that, spikes were detected using an automated Hidden Markov-Model-based algorithm for each channel (*Herbst et al., 2008*). The eye positions were obtained using an infrared-based eye-tracking device from SR Research Ltd. (Eyelink 1000 Plus). The behavioral task was designed on a standalone PC (stimulus PC) using the Psychophysics Toolbox (*Brainard, 1997*) in MATLAB (Mathworks, MA). In order to align the neural and behavioral activity (trial epochs and eye data) for data analysis, we generated strobe words denoting trial epochs and performance (rewarded or failure) during the trial. These strobe words were generated on the stimulus PC and were sent to the Plexon and Eyelink computers using the parallel port.

## Microstimulation

For arrays positioned in the prearcuate region (FEF), we used standard electrical microstimulation to confirm that saccades could be elicited with low currents. These electrodes had a depth of 5.5 mm inside the sulcus and tapered to 1 mm away from the sulcus. We conducted these microstimulation experiments after we finished our recording experiments. During the microstimulation experiment, each electrode implanted in the FEF was tested for its ability to evoke fixed-vector saccadic eye movements with stimulation at currents of 50 µA. Electrical microstimulation consisted of a 200 ms train of biphasic current pulses (1 ms, 300 Hz) with no interphase delays, delivered with a Plexon Stimulator (Plexon Inc, TX). We mapped the saccade vector elicited via microstimulation at each electrode to verify that the electrodes were implanted in the FEF. Sites at which stimulation of 50 µA or less elicited eye movements at least 50% of the time, plus regions within 2–3 mm of these locations, were considered to be in the FEF.

## Behavioral task

Each trial started with a mandatory period (500 ms) where the animal fixated on a white circle at the center of the screen. While continuing to fixate, the animal was presented with a target (a red square) for 300 ms at any one of eight locations in a 3 × 3 grid. The center square of the 3 × 3 grid contained the fixation spot and was not used. The presentation of the target was followed by a delay of 1,000 ms, during which the animal was expected to maintain fixation on the white circle at the center. At the end of this delay, a distractor (a green square) was presented for 300 ms at any one of the seven locations (other than where the target was presented). This was again followed by a delay of 1000 ms. The animal was then given a cue (the disappearance of the fixation spot) at the end of the second delay to make a saccade toward the target location that was presented earlier in the trial. Saccades to the target location within a latency of 150 ms and continued fixation at the saccade location for 200 ms was considered a correct trial. An illustration of the task is shown in *Figure 1a*. One of the animals was presented with only seven of the eight target locations because of a behavior bias in the animal.

## Cross-temporal decoding

A decoder based on linear discriminant analysis (LDA) was built using the *classify* function in MATLAB to predict the location of the target. We trained a decoder for each time point in the trial, and tested the decoder with all other time points throughout the trial. We pooled the activity across recording sessions to create a pseudo-population of 226 neurons. In the pseudo-population, for each pseudo-trial with target location *T*, we randomly picked one trial from each neuron with target

location *T,* and stacked the activity from all neurons together as if they were simultaneously recorded. We constructed 1,750 pseudo-trials (250 for each target location) as the training set, and 1,750 pseudo-trials as the testing set. The training set and testing set were sampled from non-over-lapping sets of trials from each neuron. When performing cross-temporal decoding in the full space (226 dimensions, *Figure 1b*), we denoised the training and testing data using principal components analysis (PCA) at every time point by reconstructing the data with the top *n* principal components that explained at least 95% of the variance. When performing cross-temporal decoding in the sub-space (seven dimensions, *Figure 1e,f*), the PCA projection matrix described in the previous step was replaced by the matrix specifying the desired subspace (working memory or motor preparation sub-space), and the resulting data in the subspace would thus be seven dimensional.

## Activity unmixing and subspace identification

The subspaces where identified using a pseudo-population of 226 neurons. For each trial *condition* (which was one of seven possible target locations), we trial-averaged and time-averaged the neural activity in Delay 1 (800 to 1300 ms from target onset) and Delay 2 (2000 to 2500 ms from target onset) for each neuron to obtain two activity matrices of *226 x 7*. We then normalized the two activity matrices to the mean of the baseline by subtracting neural activity in the fixation period (300 ms before target onset), and obtained activity matrices $\overline{D1}$ and $\overline{D2}$ of size *226 x 7*, where each column represented the change in population activity under one condition. After flattening $\overline{D1}$ and $\overline{D2}$ activity into 1-D arrays (each of size *1 x 1,582*, denoted as $\vec{D1}$ and $\vec{D2}$), we found high mutual information (0.33 bits) between $\vec{D1}$ and $\vec{D2}$ (*Figure 2—figure supplement 1*). We hypothesized that the highly correlated $\vec{D1}$ and $\vec{D2}$ resulted from a mixture of working memory and motor preparation activity, while working memory and motor preparation activity themselves were minimally dependent on each other. In matrix expression, we have:

$$\begin{bmatrix} \vec{D1} \\ \vec{D2} \end{bmatrix} = \begin{bmatrix} 1 & a \\ b & 1 \end{bmatrix} \begin{bmatrix} \vec{M} \\ \vec{P} \end{bmatrix}$$

where $\vec{M}$ and $\vec{P}$ are the underlying working memory and motor preparation activity in a flattened shape (each with size *1 x 1,582*); *a* and *b* are the mixing coefficients of $\vec{M}$ and $\vec{P}$. We used a standard optimization function *fmincon* in MATLAB to find out a pair of (a, b) that would recover the two activity arrays $\vec{M}$ and $\vec{P}$ with the least mutual information between them. The objective function we used was the mutual information for two discrete distributions:

$$I(X;Y) = \sum_{y=Y}\sum_{x=X} p_{(X,Y)}(x,y)\log\left(\frac{p_{(X,Y)}(x,y)}{p_X(x)p_Y(y)}\right)$$

where $p_{(X,Y)}$ is the joint probability mass function of *X* and *Y*; $p_X$ and $p_Y$ are the marginal probability mass function of *X* and *Y*, respectively. To discretize our data, we chose the number of bins according to Sturges' rule, which is conservative in estimating the number of bins (ensuring there are enough data points in each bin):

$$Number\ of\ bins = ceil(1 + log_2(N))$$

where *N* is the number of total data points from a distribution. Since a scaling operation on the two distributions would not change their mutual information, we could always stipulate the coefficient matrix to have an identity diagonal such that the interpretation of *a* and *b* would be intuitive: *a* indicated the fraction of motor preparation activity in Delay 1 if we assumed the magnitude of motor preparation activity in Delay 2 was one; *b* indicated the fraction of working memory activity in Delay 2 if we assumed the magnitude of working memory activity in Delay 1 was one. To obtain a robust optimization result, we ran the optimization function 1,000 times with random initialization and examined the results. The values for (a, b) were well centered around the valley of the objective function landscape (a = 0.118 ± 0.04, b = 0.654 ± 0.027, *Figure 2—figure supplement 1*) and the minimum mutual information obtained was 0.076 bits when we rounded *a* to 0.12 and *b* to 0.65 (*Figure 2—figure supplement 1*). We then reshaped $\vec{M}$ and $\vec{P}$ arrays into matrices with size 226 x 7 (denoted as $\overline{M}$ and $\overline{P}$, which were also referred to as two unmixed elements), and the orthonormal

bases of $\overline{M}$ and $\overline{P}$ defined the working memory and motor preparation subspaces. The column vectors in $\overline{M}$ and $\overline{P}$ were regarded as the trial-averaged and time-averaged population activity for working memory and motor preparation, respectively, for the different target locations. The working memory and motor preparation subspaces captured all target information from time-averaged data in Delay 1 and Delay 2. This was because the columns of $\overline{D1}$ and $\overline{D2}$ essentially represented the cluster means of each target location in the state space, so all the cluster means collapsed to zero in the null space of $\overline{D1}$ and $\overline{D2}$ (there was no linearly decodable information in the null space). Hence, the space spanned by $\overline{D1}$ and $\overline{D2}$ captured all target information in Delay 1 and Delay 2. Because $\overline{M}$ and $\overline{P}$ spanned the same space as $\overline{D1}$ and $\overline{D2}$, the working memory and motor preparation subspaces also captured all target information in Delay 1 and Delay 2.

A similar optimization was performed between activity in Delay 1 and the pre-saccadic period (150 to 0 ms prior to saccade onset):

$$\begin{bmatrix} \vec{D1} \\ \vec{Ds} \end{bmatrix} = \begin{bmatrix} 1 & a' \\ b' & 1 \end{bmatrix} \begin{bmatrix} \vec{M'} \\ \vec{S} \end{bmatrix}$$

where $\vec{Ds}$ was the activity in the pre-saccade period. Using the same approach as described above, we obtained $\vec{M'}$ and $\vec{S}$ with a minimum mutual information of 0.086 bits when $a' = 0.01 (\pm 0.014)$ and $b' = 0.706\ (\pm\ 0.033)$ (*Figure 3—figure supplement 2*). We reshaped $\vec{S}$ into a matrix with size *226 x 7* (denoted as $\overline{S}$), and regarded the column vectors as the trial-averaged and time-averaged pre-saccade activity.

In order to extract the unmixed working memory and motor preparation activity in the full space with single-trial variability, we used:

$$\begin{aligned} M1 &= D1 - a \times \overline{P}, \quad M2 = D2 - \overline{P} \\ P1 &= D1 - \overline{M}, \quad P2 = D2 - b \times \overline{M} \end{aligned}$$

where $\overline{M}$ and $\overline{P}$ are the unmixed trial-averaged memory and preparation activity, $D1$ and $D2$ were the single-trial Delay 1 and Delay 2 activity matrices of size *226 x 1,750* (250 random single trials per condition); $M$ and $P$ (number indicates in Delay 1 and Delay 2) were the unmixed single-trial working memory and motor preparation activity in the full space (of size *226 x 1750*). Subspace memory and preparation activity were obtained by projecting the unmixed full-space single-trial activity into their respective subspaces derived from $\overline{M}$ and $\overline{P}$.

In the error trial analysis, single-trial full-space memory and preparation activity in error trials were estimated by:

$$\begin{aligned} M1_E &= D1_E - a \times \overline{P}, \quad M2_E = D2_E - \overline{P} \\ P1_E &= D1_E - \overline{M}, \quad P2_E = D2_E - b \times \overline{M} \end{aligned}$$

where $D1_E$ and $D2_E$ were similar to $D1$ and $D2$, but from error trials. The decoder was trained and validated on the data from correct trials in the subspace and tested on the data from error trials in the same subspace. Although we interpreted the decrease in decoding performance in the two subspaces in error trials to be evidence of the link between these subspaces and the behavior of the animal, an alternative interpretation could be that there was a general increase in noise in the population in error trials (perhaps due to factors like inattention), and this led to a non-specific decrease in information in all subspaces, including the memory and preparation subspaces. In order to rule this possibility out, we quantified the intra-cluster variance in the full space across locations for correct and error trials in both Delays 1 and 2 (refer to *Figure 2—figure supplement 5*). We found no evidence supporting the fact that the intra-cluster variance in Delay 1 was higher in error trials than in correct trials ($p > 0.46$, $g = 0.85$), and found the intra-cluster variance in error trials in Delay 2 to actually be lower than correct trials ($p < 0.01$, $g = 6.6$), presumably due to the effects of divisive normalization. These results indicated that the drop in performance in the working memory and motor preparation subspaces in error trials was not due to a non-specific increase in noise, but were more likely due to the fact that the activity in error trials deviated significantly from those in correct trials, resulting in lower information in the two subspaces.

## Principal angles between subspaces

Let X and Y be two subspaces in an N dimensional full space (X has rank $x \leq N$, Y has rank $y \leq N$), and all the columns of X and Y be the orthonormal bases of each subspace. We perform a Singular Value Decomposition (SVD) on the matrix product of X and Y:

$$[U, S, V] = SVD(X^T Y)$$

where U, V are orthonormal matrices and S a diagonal matrix. The number of principal angles between X and Y is $\min(x, y)$, and these angles are obtained by computing the inverse-cosine of the diagonal elements of S and converting radians into degrees. The principal angles in all our analyses were arranged in ascending order such that the leading principal angles are more indicative of the overall alignment of two subspaces. To test if subspace X is significantly closer to subspace Y than chance (e.g. *Figure 3—figure supplement 1*), we first compute the principal angles between X and Y, and next sample the principal angles between subspace Y and 1,000 random subspaces with the same dimensionality as X to obtain $\min(x, y)$ number of chance distributions corresponding to the $\min(x, y)$ number of principal angles between X and Y. X is significantly closer to Y subspaces if the principal angels between X and Y are smaller than the 5th percentile of their corresponding chance distributions.

## Inter-to-intra cluster distance ratio

To compute the inter-to-intra cluster distance ratio for working memory activity in the full space (*Figure 2—figure supplement 5*), we bootstrapped 250 unmixed single-trial working memory activity for seven target locations. First, to compute the mean inter-cluster distance, we computed the pairwise distances between all cluster means, and then computed the grand mean of all the pairwise cluster distances. Inter-cluster distance could be intuitively understood as the measure of separation between clusters. Second, to compute the mean intra-cluster distance, we first computed the intra-cluster distance in each cluster (mean pairwise distance among all the data points in one cluster), and then computed the grand mean of the intra-cluster distance among all the clusters. Intra-cluster distance could be intuitively understood as the trial-by-trial variability among all the target conditions. The inter-to-intra cluster distance ratio is then a concept similar to the signal-to-noise ratio of the working memory activity in the state space. We repeated this procedure for 1,000 times to get a distribution of the inter-to-intra cluster distance ratio, and presented the 5th to 95th percentile of this distribution in the boxplot in *Figure 2—figure supplement 5*. The same procedure was repeated for the motor preparation activity.

To compute the inter-to-intra cluster distance ratio in subspaces (*Figure 4c,d*), we first projected the full-space activity into the desired subspaces, and then repeated the procedure mentioned above.

## Statistics

We considered two bootstrapped distributions to be significantly different if the 95th percentile range of the two distributions did not overlap. We also computed an estimated p-value for this comparison using the following formula (*Fi and Garriga, 2010*),

$$\frac{1 + X}{N + 1}$$

where $X$ represents the number of overlapping data points between the two distributions and $N$ represents the number of bootstraps. With this computation, and the $N = 1000$ bootstraps we used throughout the paper, two distributions with no overlap will result in a p-value < 0.001, and two distributions with x% of overlap will result in a p-value ~ x/100.

In addition to the estimated p-value, we also computed the effect size of the comparison using a measure known as Hedges' g, computed using the following formula (*Fisher, 1936*),

$$\left(1 - \frac{3}{4(n_1 + n_2) - 9}\right)\left(\frac{\overline{x_1} - \overline{x_2}}{s'}\right)$$

where

$$s' = \sqrt{\frac{(n_1 - 1)s_1{}^2 + (n_2 - 1)s_2{}^2}{n_1 + n_2 - 2}}$$

$\bar{x}$ refers to the mean of each distribution, n refers to the length of each distribution, and s refers to the standard deviation of each distribution.

No statistical methods were used to pre-determine sample sizes, but our sample sizes were similar to those reported in previous publications (*Murray et al., 2017*; *Stokes et al., 2013*; *Jacob and Nieder, 2014*). The majority of our analyses made use of nonparametric permutation tests, and as such, did not make assumptions regarding the distribution of the data. No randomization was used during the data collection, except in the selection of the target and distractor locations for each trial. Randomization was used extensively in the data analyzed to test for statistical significance. Data collection and analysis were not performed blind to the conditions of the experiments. No animals or data points were excluded from any of the analyses. Please see additional information in the Life Sciences Reporting Summary.

## Cell selectivity classification

For *Figure 4—figure supplement 3*, in order to match the selectivity properties of neurons in the model with those of LPFC data, we first quantified the selectivity of LPFC activity as follows. First, using a two-way ANOVA with independent variables of target locations (seven locations) and task epoch (Delay 1 and Delay 2), we categorized cells as those with *pure working memory* selectivity (those with target information in both Delay 1 and Delay 2, and those with selectivity to target location and task epoch, but no interaction, 27.6% of cells), those with *mixed* selectivity to target location and task epoch (those with a significant main effect of target location and task epoch, as well as a significant interaction between target location and task epoch, 43.9% of cells). And using two one-way ANOVAs of target location (one in Delay 1 and one in Delay 2), we categorized cells as those with *pure motor preparation* selectivity (those with significant selectivity in Delay 2, but not Delay 1, 28.6% of cells).

## Artificial neural networks

For the bump attractor model in *Figure 4—figure supplement 2*, we used two populations of firing-rate units for the memory and preparation input ($N = 80$ for each, and the whole population consisted of the working memory and motor preparation populations), and tested the model's performance with different overlapping ratios between the two populations (if the overlapping ratio was 0%, then the full network consisted of 160 units, whereas if the overlapping ratio was 100%, then the full network consisted of 80 units). The firing rate of the population was characterized by:

$$\tau \frac{dr}{dt} = -r + \varphi(W_{rec}r + W_{in}I + \sigma)$$

where $\tau$ was a uniform decay constant; $r$ was the population firing rate; $W_{rec}$ was the recurrent connection weight between units; $I$ was the external input; $W_{in}$ was the loading weight of input signal to the population; $\sigma$ was a Gaussian noise term. For numerical simulation, we used Newton's method:

$$r_{t+1} = r_t + (-r_t + \varphi(W_{rec}r_t + W_{in}I + \sigma)) \times \frac{dt}{\tau}$$

$$r_t = r_t / \alpha_t$$

where we set $\tau = 20\,ms$ and $dt = 2\,ms$; $\alpha_t$ was a scalar obtained by $mean(r_t)/mean(r_0)$, and it was applied uniformly to each unit of the whole population to maintain the mean population firing rate at the baseline level (divisive normalization). $\varphi(x)$ was a piecewise nonlinear activation function adopted from *Wimmer et al., 2014*:

$$\varphi(x) = \begin{cases} 0, & x<0 \\ x^2, & 0<x<1 \\ \sqrt{4x-3}, & x>1 \end{cases}$$

The matrix, $W_{rec}$, had a diagonal shape with stronger positive values near the diagonal, and weaker negative values elsewhere, such that only a few neighboring units were connected via excitatory weights to each other while being connected via inhibitory weights to the rest. In this way, a structured input signal to adjacent units was able to generate a local self-sustaining bump of activity. There were eight input units, representing the eight spatial target locations in the animal's task. For each input unit, the loading weight matrix $W_{in}$ specified a random group of 10 adjacent units in the working memory population, as well as the motor preparation population, to receive the signal. The input to the working memory population was transiently active in the target presentation period, and the input to the motor preparation population was transiently active in the distractor presentation period. In each trial, the label for working memory and motor preparation inputs was always the same. Distractors used the same input loadings as the working memory input did, but the strength was only 50%, and the distractor label was always different from the target label.

For the linear subspace model in *Figure 4—figure supplement 4*, we used a total of N = 112 units where the firing rate of the population was characterized by:

$$\tau \frac{dr}{dt} = -r + W_{rec}r + W_{in}I + \sigma$$

where $\tau$ was a uniform decay constant $r$ was the population firing rate; $W_{rec}$ was the recurrent connection weight between units; $I$ was the external input; $W_{in}$ was the loading weight of input signal to the population; $\sigma$ was a Gaussian noise term. For numerical simulation, we used Newton's method:

$$r_{t+1} = r_t + (-r_t + W_{rec}r_t + W_{in}I + \sigma) \times \frac{dt}{\tau}$$

$$r_t = r_t/\alpha_t$$

where we set $\tau = 20\ ms$ and $dt = 2\ ms$; $\alpha_t$ was a scalar obtained by $mean(r_t)/mean(r_0)$, and was applied uniformly to each unit of the whole population to maintain the mean population firing rate at the baseline level (divisive normalization). We constructed the recurrent weight matrix from eigendecomposition:

$$W_{rec} = Q\Lambda Q^{-1}$$

where $Q$ was a random square matrix whose columns were the eigenvectors of $W_{rec}$, and $\Lambda$ was a diagonal matrix whose diagonal elements were the corresponding eigenvalues for each eigenvector. The first 17 eigenvalues in $\Lambda$ were 1 (thus there were 17 stable eigenvectors), while the rest of the eigenvalues were randomly chosen between 0 and 1 using a uniform distribution. In a network of N neurons, the simultaneous activity of all the neurons represents a vector in an N dimensional space, and hence the vector notation and the network activity can be used interchangeably. The population activity will stay stable across time if it is a linear combination of the stable eigenvectors (*Murray et al., 2017*). In each simulation, we assigned 1 stable eigenvector as baseline activity (with entries selected from a uniform distribution U(0,1)), 8 stable eigenvectors for working memory activity, and 8 stable eigenvectors for motor preparation activity (with entries selected from U(1,2)). In order to ensure that decoding performance in Delay 1 and Delay 2 were the same, we imposed a positive mean for the motor preparation activity, so that the incorporation of motor preparation in Delay 2 would elevate the population mean, and divisive normalization would reduce the mean activity of both working memory and motor preparation information. Otherwise, if the motor preparation activity had zero mean, there would be a significant increase of decoding performance in Delay 2. In the input weight matrix, the input activity for working memory corresponded to the 8 working memory eigenvectors, and the input activity for motor preparation corresponded to the 8 motor preparation eigenvectors. For each target location, there was a one-to-one mapping of working memory activity and motor preparation activity. The distractor inputs had the same input loading as did the target inputs, but with a lower magnitude (0.2 compared to target). At the beginning of each trial, the population started with baseline activity equal to the stable baseline eigenvector, then the input for working memory was transiently active in the target presentation period, and the input for motor preparation was transiently active in the distractor presentation period. When the input activity had the same direction as a stable eigenvector, the resultant population activity would stay stable across

time because it was still a linear combination of stable eigenvectors. As all the input activity corresponded to stable eigenvectors, all target information, distractor information, and motor preparation information were maintained stably across time.

## Subspaces for uncorrelated information

Due to our experimental design, the working memory location and the motor preparation locations were identical in each trial, and thus correlated. We can imagine another case where there are two types of information that do not have a one-to-one mapping (for example, in a task that requires memorizing locations of items - one out of two possible locations, and their colors - one out of three possible colors, which are uncorrelated). When each target location is associated with only one stimulus color (similar to our working memory and motor preparation locations), the incorporation of stimulus color information in Delay 2 would add only one out of three possible shifts (representing the three possible stimulus colors) to the clusters representing target location (*Figure 2—figure supplement 7a*). However, when target location and stimulus color are uncorrelated (each stimulus color is equally likely to appear in each target location), the incorporation of stimulus color information could add any of the three possible shifts to the clusters representing target location activity, leading to much more diffuse clusters (*Figure 2—figure supplement 7b*). In this latter case, we propose a more general formulation to estimate the information subspaces for target location and stimulus color. First, we group trials by target location and obtain the trial-averaged and time-averaged activity in Delay 1 ($\overline{G_1}$). Next, we group trials by stimulus color and obtain the trial-averaged and time-averaged activity in Delay 2 ($\overline{G_2}$). Finally, we estimate the subspaces by:

$$\begin{aligned} \overline{G_1} &= \overline{L} + a \times E\left[\overline{C_L}\right] \\ \overline{G_2} &= \overline{C} + b \times E\left[\overline{L_C}\right] \end{aligned}$$

where $\overline{L}$ and $\overline{C}$ define the subspaces for target location and stimulus color, while $a$ and $b$ are scalars representing the mixing coefficients. $E\left[\overline{C_L}\right]$ represents the expectation of stimulus color activity associated with particular target locations, and $E\left[\overline{L_C}\right]$ represents the expectation of target location activity associated with particular stimulus colors. At one extreme, the correlation between target location and stimulus color could be 0 (completely random pairing), in which case the expectation value will reduce to 0 if averaging across the other variable does not result in a net translation (which also means there will be no code morphing). In this case, there is no need to minimize mutual information, as the $\overline{L}$ and $\overline{C}$ vectors will remain unchanged. On the other hand, if there is a net translation, code morphing will be present, and there will be a need to minimize the mutual information to recover the angles between the subspaces. At the other extreme, the correlation could be 1 (one-to-one mapping), in which case the expectation value will reduce to $\overline{C}$ and $\overline{L}$. We can verify that the decorrelation method used for working memory and motor preparation elements in this work was a special case of this formulation ($\overline{G_1}$ and $\overline{G_2}$ become $\overline{D_1}$ and $\overline{D_2}$ as memory and preparation have the same grouping, and the expectations reduce to $\overline{P}$ and $\overline{M}$ as there is one-to-one mapping). We would perform the same optimization on $(a, b)$ that will give the least mutual information between $\overline{L}$ and $\overline{C}$.

## Data and code availability statement

The code package and data needed to perform the analyses used in the paper is available at https://github.com/chengtang827/MemoryPreparationSubspace (*Tang, 2020*; copy archived at https://github.com/elifesciences-publications/MemoryPreparationSubspace).

## Acknowledgements

This work was supported by startup grants from the Ministry of Education Tier 1 Academic Research Fund and SINAPSE to CL, a grant from the NUS-NUHS Memory Networks Program to S-CY, a grant from the Ministry of Education Tier 2 Academic Research Fund to CL and S-CY (MOE2016-T2-2-117), and a grant from the Ministry of Education Tier 3 Academic Research Fund to CL and S-CY (MOE2017-T3-1-002).

## Additional information

### Funding

| Funder | Grant reference number | Author |
|---|---|---|
| Ministry of Education - Singapore | MOE2016-T2-2-117 | Camilo Libedinsky<br>Shih-Cheng Yen |
| Ministry of Education - Singapore | MOE2017-T3-1-002 | Camilo Libedinsky<br>Shih-Cheng Yen |
| National University of Singapore and National University Health System Strategic Research Award | | Shih-Cheng Yen |

The funders had no role in study design, data collection and interpretation, or the decision to submit the work for publication.

### Author contributions
Cheng Tang, Conceptualization, Data curation, Software, Formal analysis, Investigation, Visualization, Methodology, Writing - original draft, Writing - review and editing; Roger Herikstad, Conceptualization, Software, Investigation, Methodology; Aishwarya Parthasarathy, Conceptualization, Data curation, Investigation, Methodology; Camilo Libedinsky, Shih-Cheng Yen, Conceptualization, Resources, Supervision, Validation, Investigation, Methodology, Writing - original draft, Project administration, Writing - review and editing

### Author ORCIDs
Cheng Tang https://orcid.org/0000-0002-5183-1387
Shih-Cheng Yen https://orcid.org/0000-0001-7723-0072

### Ethics
Animal experimentation: All animal procedures were approved by, and conducted in compliance with the standards of the Agri-Food and Veterinary Authority of Singapore and the Singapore Health Services Institutional Animal Care and Use Committee (SingHealth IACUC #2012/SHS/757). The procedures also conformed to the recommendations described in Guidelines for the Care and Use of Mammals in Neuroscience and Behavioral Research (National Academies Press, 2003).

### Decision letter and Author response
Decision letter https://doi.org/10.7554/eLife.58154.sa1
Author response https://doi.org/10.7554/eLife.58154.sa2

## Additional files

### Supplementary files
• Transparent reporting form

### Data availability
The code package and data needed to perform the analyses used in the paper is available at: https://github.com/chengtang827/MemoryPreparationSubspace (copy archived at https://github.com/elifesciences-publications/MemoryPreparationSubspace).

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
