## [Decision Letter]

Thank you for submitting your article "Minimally dependent activity subspaces for working memory and motor preparation in the lateral prefrontal cortex" for consideration by *eLife*. Your article has been reviewed by two peer reviewers, and the evaluation has been overseen by a Reviewing Editor and Michael Frank as the Senior Editor. The reviewers have opted to remain anonymous.

The reviewers have discussed the reviews with one another and the Reviewing Editor has drafted this decision to help you prepare a revised submission.

Summary:

After consultation, the two reviewers and I wish to highlight the following points that will need to be addressed (along with any other issues raised in the reviews below).

1) Were the data pooled across FEF and 9/46? FEF and 9/46 cytoarchitecture is different and they may work as interrelated but somewhat independent modules (e.g. FEF is an eye field and 9/46 is not based on the work of Schiller, Tehovnik, Goldberg, Schall, etc). To assess the potentially different roles of these areas, it will be important to perform a separate analysis of the data per area, as well as a more complete presentation of the data, anatomy, and recording sites.

2) Clarity of the data analysis method. The authors are using a novel method for data analysis and some more effort is needed to explain concepts more carefully and coherently. See reviewer comments for more specifics.

3) We all agree that there are some issues with the model design and assumptions that limit the insights gained. Given that we are requesting more information about FEF/9-46 and about recording sites etc, we suggest that the model, after revising, might be moved to the supplementary materials.

4) The working memory and motor preparation functions alluded to in the manuscript are very difficult to separate given the design of this experiment (see reviews for more details). An analysis of error trials may help to address this issue, as has been done in some rodent studies (e.g., work from Karel Svoboda, as in Inagaki et al., 2018).

Reviewer #1:

This manuscript examines the coding of working memory signals in two areas of the macaque frontal cortex, the Frontal Eye Fields and the Lateral Prefrontal Cortex. The paper concludes that there are two different subspaces in the DLPFC, one encoding WM signals and the other motor preparation signals, and that neurons with mixed selectivity contribute to both subspaces. They conduct an analysis of multielectrode data recordings in monkeys performing an ODR task with an intervening distractor. This is an interesting study that follows up on two previous publications by the same group (Parthasarathy et al., 2017 and 2019). The study has potential to make a contribution to our knowledge or how worming memory and movement signals segregate within the frontal lobe. Below I will provide a summary of what I believe are strengths and weaknesses of the current manuscript and suggestion on how, in my opinion, the manuscript could be improved.

Strengths:

1) The paper conducts and elegant and novel analysis of multielectrode data recordings using a standard working memory task (ODR), to which a distracter has been added. The methods are state-of-the-art and this reviewer acknowledges the difficulties of conducting the experiments and achieving good isolation of units with relatively large amount of electrodes implanted in the cortex. These procedures are challenging, and the authors should be praised for that.

2) The idea of different subspaces encoding working memory signals and motor preparation within the LPFC is interesting and controversial. Neurons in the LPFC have been hypothesized encode representations that are dissociated from motor signals (Funahashi et al., 1993). However, studies in the same area have also reported neurons that respond before a saccadic eye movement is made (Bullock et al., 2017). The engagement of the LPFC in motor preparation is not clear, mainly because must studies do not dissociate the location of the saccade and the relevant task location (but see Lebedev et al., 2004). In the latter study, however, there was again no dissociation of attention and movement goal. On the other hand, neurons in the neighboring to FEF and more posterior premotor cortex do encode movement plans (Cisek and Kalaska., 2005, see their Figure 5 for recording locations). Regarding neurons in the FEF there is also a consensus they encode movement goals (See Tehovnik, 2000 for a review and Vernet et al., 2014). To summarize this is an interesting issue that has not been clarified yet. Would LPFC encode motor preparation signals rather than abstract representations of space?

Weaknesses and suggestions:

1) One main issue is that the authors pool together the data from FEF and the LPFC. These are two different brain areas and it has been shown that although some neurons exhibit similar properties they are not the same. Neurons in the FEF have a more direct connection to motor centers (e.g., stimulation of the FEF with low current intensities <50microAmps produces saccades, see also Tehovnik, 2000 for a review). This is not the case in LPFC. One issue is that the area of the LPFC where recordings were conducted is not described in the paper, although presumably this is the same data set as Parthasarathy et al., 2017 where there is a figure describing the location of the arrays. By examining the data in the previous publication by the same group (Parthasarathy et al., 2017) my guess the LPFC areas are 9/46, in the vicinity of the principal sulcus (Petrides, 2005). This needs to be detailed in the manuscript. From the point of view that neurons in FEF and 9/46 share some properties, it may make sense pooling the data. However, from the anatomical and other physiological properties point of view it is not justified. Something the authors should take into account is the FEF is not considered by many as part of the granular prefrontal cortex. The cytoarchitecture of the FEF is disgranular, not granular, as areas 9/46 (see Figure 1 of Petrides, 2005). The FEF also has large pyramidal neurons in layer V (Stanton et al., 1989), which is different from areas 9/46, where the largest pyramidal cell bodies are in layers 2/3. The connectivity with other areas is also different (see below). A comprehensive review of FEF anatomy and function across species is in Vernet et al. 2014. Although the review of Tehovnik, 2000, used here as a reference seems to allocate FEF to the DLPFC, this does not seem compatible with the structure of the area in terms of granularity (Petrides, 2005). Other reviews such as Thompson, 2005 make the distinction between FEF and the area around the principal sulcus (9/46). The latter is the area where Goldman-Rakic and coworkers conducted many studies of working memory and I believe some of the arrays in this study were implanted. What I am trying to get at is that FEF does not have the same features as areas 46/9, and therefore it justifies separate analyses rather than pooling the data. It is hard to conceive that a read-out mechanism is using data from these regions simultaneously, not impossible but it would not be the first assumption to make. I would suggest analyzing the data from FEF and areas 9/46 separately. This may reveal that the premotor subspace is biased towards the FEF and the memory towards areas 9/46.

2) One issue in this dataset is the fact that the task does not dissociate the memorized location and the location of the saccade. This is problematic because one needs to assume the memory for the location ends when the saccade starts, some sort of sequential order in the task. This may be the case but there is data from Funahashi and Goldman-Rakic in which they use an antisaccade task and show that the LPFC neurons in areas 46/9 encode the remembered location rather than the location of the saccade (Funahashi et al., 1993). This suggest that motor codes, or preparatory motor signals for saccades may not be encoded in the LPFC. In the FEF this is very different, as commented above, the FEF seems to be directly connected to oculomotor centers such as the SC (see Hanes and Wurtz, 2001). I am not aware of similar connectivity between areas 9/46 and SC, at least not to the same degree, which is relevant to the ODR task employed in this study (see Field et al., 2008). The FEF seems to be part of an oculomotor network while 46/9 are not. This brings me to suggesting the authors again to perform separate analyses for the FEF and LPFC data and to include a diagram in the paper that shows the recording locations superimposed to an anatomical map (e.g., Petrides, 2005).

3) The existence of a distracter in the middle of the delay period is problematic in this task. The animals are supposed to ignore the distracter and they did so as evidenced in the performance data. One question regarding the distracter in the task is whether the second subspace the authors find is encoding a memory for the distracter that diverges from the code in the first subspace. It has been shown that neurons in the LPFC encode visual/perceptual and memory signals and that this can be different populations (Mendoza-Halliday et al., 2017). One possibility is that the authors group the trials according to the distracter location and repeat the analysis. One would anticipate some neurons in the population respond to the distracter. Whether the animal is aware of the distracter location at the time of the saccade is impossible to know for sure. However, if one analyses the pattern of errors and one sees a bias making saccades to the distracter location that would suggest there was a memory for that component of the task that it may be maintained together with the one for the target or in some cases deleted the memory for the target. The authors seem to be aware of the issues with the distracter but they concentrate on color rather than location. Color signals are encoded in the LPFC (Schwedhelm et al., 2020), making the pooling difficult if the authors had several colors of the distracter. But again, pooling by distracter location should be doable. I suggest repeating the analyses as a function of distracter location and examine whether a subspace may encode the distracter location. This would not be surprising, working memory can hold more than 1 item at the time.

4) The paper contains few main figures with very little information about the recordings sites, responses of single neurons. It is heavy on the analysis and data modeling side but does not show single cell or population data visualizations. I am not sure if the authors consider this or they show these data in the previous manuscript. But this should be an independent manuscript. For a modeling paper this may be fine but this is not my view of this manuscript. In this case it is critical to show characterization of responses of single units, levels of firing rates, examples, analysis of selectivities for remembered location, saccade location, etc. The paper could be much more appealing if incorporating these suggestions.

Reviewer #2:

The work by Tang and colleagues pertains to the identification of neural subspaces that are computationally relevant to a task that requires representation of a target across two delay periods, robustness to distractors, and motor execution to the initially acquired target. It builds upon previous work that showed “code morphing” of working memory representations during a delay period following distractor presentation (Parathasurathy et al., 2017). A notable aspect of the present study is the description of an optimization algorithm to identify neural subspaces that does not rely on “classical” dimensionality reduction techniques such as orthogonal decomposition (e.g., PCA, dPCA, etc). The authors claim that the algorithm leads to the existence of two distinct yet concurrent cognitive subspaces within the same task and period: a working memory and a motor preparation subspace. In principle, these results constitute a valuable addition to the literature on population analyses as well as on the interaction between memory and motor processes. The lack of clarity of the presentation of the method, however, obscures the overall interpretation of the data that supports the authors' claims, while making the description challenging for the reader. With respect to the computational model, many important details require revision and/or clarification. Moreover, the model limits itself to reproducing decoding performance, but says much less about the biophysical mechanisms that underlie the existence and/or deployment of working memory and motor preparation subspaces. Although I commend the authors on relating population activity patterns to neural circuit mechanisms, the model in its current form is not particularly insightful.

Below I provide a detailed summary of my three main concerns. The first two points (“Details of subspace identification method” and “Modeling”) are critical:

1) Details of subspace identification method:

a) Rigor and consistency in definitions:

Subsection “Two minimally dependent subspaces coexisted within the LPFC”: Several technical terms related to matrix decomposition are introduced but many of them are used ambiguously throughout, and there is conceptual overlap between them: activity, unmixing matrix, component, and subspace.

“…the magnitude of one component in Delay 2 was 65% of that in Delay 1 (Component 1), and the magnitude of the other component in Delay 1 was 12% of that in Delay 2 (Component 2).”: Components are conflated with their magnitude

“The temporal dynamics of the full space population activity projected into these subspaces showed that activity in the first subspace emerged early after target presentation and was maintained until the saccade cue…”: Projections or magnitude of the projections?

In what sense is the term “information” used throughout the manuscript? (In the vernacular sense or as a measure of uncertainty in bits?)

“…information emerged right after target presentation, and although the information was stronger in Delay 1 (60.5 ± 1.3%),”: What is the relationship between target information and decoder performance? are both variables measured with a percentage?

“Cross-temporal decoding of full space neural activity projected into the second subspace showed that information emerged after distractor presentation (42.6 ± 1.1%), and was stable throughout Delay 2 (Figure 1F). In Delay 1 and Delay 2, the first subspace explained 14.6% and 10.3% of the full space variance, while the second subspace explained 5.8% and 8.1% of the full space variance, respectively.”: What is the relationship here between the variance explained by subspaces and the components? The distinction between “component” and “subspace” is not always clear.

“…and the first 6 out of the 7 principal components cumulatively accounted…”: In standard dimensionality reduction, for example, the goal is to reduce the dimensionality of the neuron state space (226 neurons) to a lower number (e.g., Santhanam et al., 2009). What do the 6 effective dimensions refer to?

Subsection “Information in one subspace led to a small amount of interference in information in the other subspace”, “We found no evidence of a drop in performance between proj_MSub_(M) and proj_MSub_(M+P) (p > 0.73, g = 0.61), and between proj_PSub_(P) and proj_PSub_(M+P) (p > 0.22, g = 2.63), suggesting a lack of interference between these subspaces”: How is this result interpreted in light of the finding that the subspaces are not orthogonal?

b) Novelty and relationship to established methods:

Subsection “Two minimally dependent subspaces coexisted within the LPFC”: There is a discussion of the difficulty of obtaining labels for working memory and motor preparation subspaces, and the subsequent application of a novel method to obtain them. How is this different from orthogonalization? In other words, how is considering mutual information between components a more general framework?

Aforementioned subsection, “Our objective was then to find through an optimization technique the best unmixing matrix, to apply to Delay 1 and Delay 2 activity, that could recover the working memory and motor preparation activity with the lowest mutual information possible between them (see Materials and methods).”: How is this different from the optimization method used in Parathasurathy et al., 2019? Is this a general framework to identify any pair of subspaces?

“This was qualitatively consistent with our hypothesis, aside from the decrease in information in Delay 2 (39.9 ± 1.1%).”: Is the decrease in “information” in Delay 2 related to the fact that this subspace is morphed in Delay 2 (Parathasarathy et al., 2017)?

“Our results provided the first evidence showing that information about two separate cognitive processes can be simultaneously encoded in subspaces within the same brain region.”: This is a strong statement for at least three reasons: (a) there are other studies that have shown some decomposition related to what can arguably be called cognitive processes, including some from the rodent literature which were not referenced (e.g., Inagaki et al., 2019, Svoboda and Li, 2018); (b) this work has not conclusively/causally shown that these subspaces indeed contain information about memory and motor preparation, as lesion studies would; and (c) the word subspace in this study does not imply reduced dimensionality as in many other studies (e.g. Kaufman et al., 2014). I suggest rewriting in a more modest form.

c) Overall clarity in description:

A method is described to obtain working memory and motor preparation subspaces. However, the logical presentation is not adequate. In Figure 1, for example, subspaces are already introduced to be related to working memory and motor preparation, without any evidence. An alternative is to name the two subspaces in Figure 1 only after showing that they correspond to functional subspaces (before they are, Subspace 1 and Subspace 2).

“Since the second subspace contained target information only after the distractor disappeared, and motor preparation presumably began after the last sensory cue that reliably predicted the timing of the Go cue (i.e. the offset of the distractor), we hypothesized that the second subspace corresponded to a motor preparation subspace”: Can motor preparation and working memory be distinguished in terms of behavior during error trials (e.g., Inagaki et al., 2018)?

“In Delay 1 and Delay 2, the first subspace explained 14.6% and 10.3% of the full space variance, while the second subspace explained 5.8% and 8.1% of the full space variance, respectively.”: This sentence about explained variance is difficult to parse, because of the existence of two delays, two subspaces and the full space. What would be the expected total variance? Are the subspaces explaining a fraction of the variance of the full space?

2) Modeling

a) model construction:

i) Equation in subsection “Model” describes the dynamics of the “bump attractor” model. A “leak” term, which reflects a decay towards a baseline firing rate, seems to be missing. From a modeling point of view, the leak is crucial for the dynamics, as its existence is what motivated recurrent synaptic circuitry to counterbalance the decay mediated by the leak for integration and working memory computations as mediated by “bump attractor” dynamics (e.g., compare to Wimmer et al., 2014, which was cited).

ii) There are no details of how “normalization” was implemented. Was it applied to all units uniformly? What is the form of the normalization equation and what are the values of the parameters? It seems like an ad-hoc step, while some models have shown that normalization arises naturally in some implementations of the bump attractor model (e.g., Ardid et al., 2007).

iii) Subsection “Model” paragraph two: The description of the inputs here (i.e., working memory and motor preparation) is missing some details. Are the inputs correlated? How does the distractor input affect the overall dynamics? What are their values?

iv) In subsection “A bump attractor artificial neural network with divisive normalization recapitulated the properties of LPFC activity” there is a description of how the model is constrained by selectivity of the neural data, as well as by the “ 43 % overlap” in inputs (Figure 5). How this related to the loadings found in Figure 3?

b) Model significance:

i) What are the activities of the model neurons? In particular, it is not clear whether the neurons are mixed selective, i.e., whether any given (simulated) neuron participates in both working memory and motor preparation. A hint is given in Figure 4—figure supplement 2, but this should be expanded.

ii) In Figure 5, a direct comparison between model and data is missing, although it is described qualitatively in the text.

iii) What are the model predictions?

3) Additional analyses or plots to improve clarity

a) Is Figure 1C necessary, given a similar Figure 1G with actual data? A general schematic is always welcome, perhaps illustrating distinct possibilities of overlap between the subspaces.

b) “In an additional test of the hypothesis that the second subspace corresponded to a motor preparation subspace, we examined the relationship between the unmixed motor preparation activity and the unmixed pre-saccade activity at the level of single cells”: This is a continuation, at a single cell level, of the previous hypothesis of relating delay 2 period activity to pre-saccadic activity. For an independent analysis, can the activity in error trials be used to distinguish motor preparation from working memory representations?

c) “Importantly, Component 2 and Component 2’ were also significantly correlated (Figure 2A right, Pearson correlation r = 0.62, P <0.01).”: Can the similarity between the components be defined in terms of angles, as done between the subspaces? Moreover, the correlation component2-component2' = 0.62 is high, but could be higher. How does the motor preparation subspace relate to distractor encoding? Could this explain why the correlation 2-2' is not higher?

d) “…concurrent increase of activity in Neurons 1 and 2 signals a change in memory (i.e. the population activity in state space moved along the working memory subspace)…”: This has not been shown explicitly with the neural activity, namely, that moving along the working memory subspace, corresponds to a (discrete) change in the target representation. Such a figure would be much more informative than the illustration in Figure 3C (which is appreciated, but would actually be valid for any orthogonal subspace decomposition, not necessarily working memory/motor preparation).

---

## [Author Response]

Summary:After consultation, the two reviewers and I wish to highlight the following points that will need to be addressed (along with any other issues raised in the reviews below).1) Were the data pooled across FEF and 9/46? FEF and 9/46 cytoarchitecture is different and they may work as interrelated but somewhat independent modules (e.g. FEF is an eye field and 9/46 is not based on the work of Schiller, Tehovnik, Goldberg, Schall, etc). To assess the potentially different roles of these areas, it will be important to perform a separate analysis of the data per area, as well as a more complete presentation of the data, anatomy, and recording sites.

We have clarified the recording sites used in this current analysis in the main text and in the response to reviewers (see below for more details). In summary, we only used neurons recorded from electrodes in area 9/46 for this analysis, since cells recorded in those electrodes were the ones that showed a higher degree of mixed selectivity and code morphing (in Parthasarathy et al., 2017). We did not include neurons in the FEF (which were confirmed using microstimulation in Parthasarathy et al., 2017), as they exhibited a stable code that implied a single subspace.

2) Clarity of the data analysis method. The authors are using a novel method for data analysis and some more effort is needed to explain concepts more carefully and coherently. See reviewer comments for more specifics.

We realize that the methods were not explained clearly enough, so we have modified the explanation extensively (as explained below in the response to reviewers and as reflected in the changes to the main text of the manuscript). We hope that the changes make it clearer why we needed a new method, and what this method entailed.

3) We all agree that there are some issues with the model design and assumptions that limit the insights gained. Given that we are requesting more information about FEF/9-46 and about recording sites etc, we suggest that the model, after revising, might be moved to the supplementary materials.

We agree with this point, and have moved the model to the supplement, as suggested.

4) The working memory and motor preparation functions alluded to in the manuscript are very difficult to separate given the design of this experiment (see reviews for more details). An analysis of error trials may help to address this issue, as has been done in some rodent studies (e.g., work from Karel Svoboda, as in Inagaki et al., 2018).

It would have been great to have more error trials to be able to carry out this analysis. Unfortunately, the elevated performance of our monkeys prevented us from obtaining enough error trials to run a meaningful analysis. Furthermore, even if we did have enough error trials, the suggested analysis would be challenging, since an error could be due to failures in memory encoding and/or motor preparation. We expand on these points in the response to reviewers below.

Reviewer #1:[…]Weaknesses and suggestions:1) One main issue is that the authors pool together the data from FEF and the LPFC. These are two different brain areas and it has been shown that although some neurons exhibit similar properties they are not the same. Neurons in the FEF have a more direct connection to motor centers (e.g., stimulation of the FEF with low current intensities <50microAmps produces saccades, see also Tehovnik, 2000 for a review). This is not the case in LPFC. One issue is that the area of the LPFC where recordings were conducted is not described in the paper, although presumably this is the same data set as Parthasarathy et al., 2017 where there is a figure describing the location of the arrays. By examining the data in the previous publication by the same group (Parthasarathy et al., 2017) my guess the LPFC areas are 9/46, in the vicinity of the principal sulcus (Petrides, 2005). This needs to be detailed in the manuscript. From the point of view that neurons in FEF and 9/46 share some properties, it may make sense pooling the data. However, from the anatomical and other physiological properties point of view it is not justified. Something the authors should take into account is the FEF is not considered by many as part of the granular prefrontal cortex. The cytoarchitecture of the FEF is disgranular, not granular, as areas 9/46 (see Figure 1 of Petrides, 2005). The FEF also has large pyramidal neurons in layer V (Stanton et al., 1989), which is different from areas 9/46, where the largest pyramidal cell bodies are in layers 2/3. The connectivity with other areas is also different (see below). A comprehensive review of FEF anatomy and function across species is in Vernet et al., 2014. Although the review of Tehovnik, 2000, used here as a reference seems to allocate FEF to the DLPFC, this does not seem compatible with the structure of the area in terms of granularity (Petrides, 2005). Other reviews such as Thompson, 2005 make the distinction between FEF and the area around the principal sulcus (9/46). The latter is the area where Goldman-Rakic and coworkers conducted many studies of working memory and I believe some of the arrays in this study were implanted. What I am trying to get at is that FEF does not have the same features as areas 46/9, and therefore it justifies separate analyses rather than pooling the data. It is hard to conceive that a read-out mechanism is using data from these regions simultaneously, not impossible but it would not be the first assumption to make. I would suggest analyzing the data from FEF and areas 9/46 separately. This may reveal that the premotor subspace is biased towards the FEF and the memory towards areas 9/46.

We thank the reviewer for pointing out that we provided incomplete information about our recording sites and neural data involved in the analyses. We did not pool together FEF and LPFC neurons in our analyses; the data from what we called LPFC in this paper were only from Areas 46 and 9/46, and did not include what we previously identified as FEF in Parthasarathy et al., 2017.

We have added a new figure (Figure 1B) to show the recording sites of the LPFC cells, and highlight the difference between LPFC and FEF electrode positions.

We have added a sentence in Results to state that the data used in the analyses did not involve FEF neurons:

“Figure 1B shows the different electrode positions in the LPFC and FEF on an anatomical map. Additionally, FEF electrodes were differentiated from LPFC electrodes using microstimulation (see Materials and methods).”

We have also added a section in Materials and methods to describe how we functionally differentiated the FEF from the LPFC using microstimulation.

In Parthasarathy et al., 2017, one of the main findings was that LPFC neurons exhibited code-morphing (i.e. there were two different population codes in Delay 1 and Delay 2 for the same target location), whereas the population code in FEF did not morph (i.e. there was one stable code throughout Delay 1 and Delay 2). This motivated us to use only LPFC data, rather than FEF data, to identify the two subspaces in the current work.

We have added a line in Results to highlight this motivation.

“In this paper, the presence of code-morphing in the LPFC motivated us to analyze the 226 single neurons recorded from the LPFC, which did not include those recorded from the FEF.”

2) One issue in this data set is the fact that the task does not dissociate the memorized location and the location of the saccade. This is problematic because one needs to assume the memory for the location ends when the saccade starts, some sort of sequential order in the task. This may be the case but there is data from Funahashi and Goldman-Rakic in which they use an antisaccade task and show that the LPFC neurons in areas 46/9 encode the remembered location rather than the location of the saccade (Funahashi et al., 1993). This suggest that motor codes, or preparatory motor signals for saccades may not be encoded in the LPFC. In the FEF this is very different, as commented above, the FEF seems to be directly connected to oculomotor centers such as the SC (see Hanes and Wurtz, 2001). I am not aware of similar connectivity between areas 9/46 and SC, at least not to the same degree, which is relevant to the ODR task employed in this study (see Field et al., 2008). The FEF seems to be part of an oculomotor network while 46/9 are not. This brings me to suggesting the authors again to perform separate analyses for the FEF and LPFC data and to include a diagram in the paper that shows the recording locations superimposed to an anatomical map (e.g., Petrides, 2005).

We thank the reviewer for raising two good points in this comment, which we address below.

First, while it is true that the task does not dissociate the memorized location and the location of the saccade (which is the reason why we developed this novel method to identify the 2 subspaces), it is not the case that we required a sequential order in the task, as working memory and motor preparation information overlapped in Delay 2. Our interpretation of the 2nd subspace as a motor preparation subspace is consistent with the interpretation that the monkeys begin the preparation of the movement after the last stimulus that reliably predicts the Go cue (which is the offset of the distractor, that reliably occurs 1 second prior to the Go cue). But it is worth highlighting that this is our interpretation of the results, rather than a guiding hypothesis. In other words, our method to identify the 2 subspaces included no constraint or bias towards finding a subspace where information emerged in Delay 2. Rather, the fact that we did find a subspace with information that emerged during Delay 2 led us to interpret this subspace as a motor preparation subspace.

Second, the concern in Point 2 stems from the same problem raised in the Point 1. Following the reviewer’s suggestion, we have added a diagram to visually show the electrode locations on an anatomical map, and stated explicitly that all our analyses with LPFC neurons was not mixed with FEF neurons (please refer to the responses to Point 1).

3) The existence of a distracter in the middle of the delay period is problematic in this task. The animals are supposed to ignore the distracter and they did so as evidenced in the performance data. One question regarding the distracter in the task is whether the second subspace the authors find is encoding a memory for the distracter that diverges from the code in the first subspace. It has been shown that neurons in the LPFC encode visual/perceptual and memory signals and that this can be different populations (Mendoza-Halliday et al., 2017). One possibility is that the authors group the trials according to the distracter location and repeat the analysis. One would anticipate some neurons in the population respond to the distracter. Whether the animal is aware of the distracter location at the time of the saccade is impossible to know for sure. However, if one analyses the pattern of errors and one sees a bias making saccades to the distracter location that would suggest there was a memory for that component of the task that it may be maintained together with the one for the target or in some cases deleted the memory for the target. The authors seem to be aware of the issues with the distracter but they concentrate on color rather than location. Color signals are encoded in the LPFC (Schwedhelm et al., 2020), making the pooling difficult if the authors had several colors of the distracter. But again, pooling by distracter location should be doable. I suggest repeating the analyses as a function of distracter location and examine whether a subspace may encode the distracter location. This would not be surprising, working memory can hold more than 1 item at the time.

We didn’t find evidence that the animals are more prone to make an error saccade to the distractor location, as shown in Author response image 1.

**Author response image 1. sa2fig1:** Each point represents the probability of making a saccade into the distractor location in a single session. Blue line, chance probability (1/8). No significant evidence was found against chance (T-test, P > 0.31).

Although there was no tendency to saccade to the distractor location, there was still significant information for distractor locations in the population activity (Figure 3—figure supplement 3A). But the distractor information was not encoded in the motor preparation subspace, as we have shown that the decoding performance for distractor locations in the motor preparation subspace was significantly lower than that in the full space (Figure 3—figure supplement 3B).

We have added this supplementary figure and a section in Results to clarify this point.

“Alongside the working memory and motor preparation activities for target locations, there could also be activities representing distractor locations in Delay 2. By grouping trials according to distractor labels, we indeed found significant distractor information in the full space (Figure 3—figure supplement 3). However, the distractor activity in Delay 2 was not related to the *Element 2* or the motor preparation subspace we identified, because the distractor activity and the motor preparation activity were obtained from data grouped by different trial labels (target and distractor labels were uncorrelated). Very little distractor information (17.9 ± 0.7%) was successfully decoded in the motor preparation subspace (Figure 3—figure supplement 3).”

4) The paper contains few main figures with very little information about the recordings sites, responses of single neurons. It is heavy on the analysis and data modeling side but does not show single cell or population data visualizations. I am not sure if the authors consider this or they show these data in the previous manuscript. But this should be an independent manuscript. For a modeling paper this may be fine but this is not my view of this manuscript. In this case it is critical to show characterization of responses of single units, levels of firing rates, examples, analysis of selectivities for remembered location, saccade location, etc. The paper could be much more appealing if incorporating these suggestions.

We agree with the reviewer that these changes will make the manuscript more readable and appealing. Together with the diagram showing the anatomical location of electrodes (Figure 1B), we added diagrams showing two single cell responses (Figure 1C) and a characterization of the population selectivity (Figure 1D) to form a new Figure 1.

We also changed our Results section accordingly:

“We recorded single unit activity from the LPFC and FEF of both monkeys while they performed the task. Figure 1B shows the different electrode positions in the LPFC and FEF on an anatomical map. Additionally, FEF electrodes were differentiated from LPFC electrodes using microstimulation (see Materials and methods).”

“In this paper, the presence of code-morphing in the LPFC motivated us to analyze the 226 single neurons recorded from the LPFC, which did not include those recorded from the FEF. Single neurons in the LPFC showed sustained selectivity to target locations during both delay periods, with some maintaining the same target tuning in both delays (Figure 1C, left), while some changed target tuning from Delay 1 to Delay 2 (Figure 1C, right). The latter category of neurons was characterized as non-linearly mixed selective neurons, and was shown to drive code-morphing in LPFC. On the population level, most of the cells with target selectivity in one delay also showed selectivity in the other delay (Figure 1D).”

Reviewer #2:[…]1) Details of subspace identification method:a) Rigor and consistency in definitions:Subsection “Two minimally dependent subspaces coexisted within the LPFC”: Several technical terms related to matrix decomposition are introduced but many of them are used ambiguously throughout, and there is conceptual overlap between them: activity, unmixing matrix, component, and subspace.

We thank the reviewer for the suggestions to enhance the clarity and readability of this manuscript.

We have chosen to use “activity” to refer to the raw firing rate of neurons (spikes/s), such as Delay 1 or Delay 2 activity; it is also used to refer to the contribution that the firing rate makes to working memory (working memory activity) or motor preparation (motor preparation activity).

We used the term “unmixing matrix” to refer to the matrix of coefficients used to unmix the original Delay 1 and Delay 2 activity. We have renamed it to “unmixing coefficients”.

We used “component” to refer to the seven 226-dimensional vectors obtained from the optimization. To avoid confusion with the meaning of “component” in other technical contexts like principal component analysis, we have changed it to “element”, and explicitly defined an “element” as the 7 vectors obtained by the optimization. When talking about “elements”, we are always referring to the 226 x 7 vector sets.

A “subspace” is the orthonormal bases of an “element” (7 vectors). We have also added this definition.

We have modified the text as follows:

“Using our method, we started with Delay 1 and Delay 2 activity exhibiting 0.33 bits mutual information, and found two unmixed elements (each of size 226 x 7) from D1 and D2 activity with a minimum mutual information of 0.08 bits (Figure 2—figure supplement 1). The two elements we identified consisted of 7 vectors in the 226-dimensional space, and according to the unmixing coefficients we identified, the magnitude of one element (*Element 1*) in Delay 2 was 65% of that in Delay 1, and the magnitude of the other element (*Element 2*) in Delay 1 was 12% of that in Delay 2. The orthonormal bases of the two elements defined two subspaces (*Subspace 1* and *Subspace 2*).”

“…the magnitude of one component in Delay 2 was 65% of that in Delay 1 (Component 1), and the magnitude of the other component in Delay 1 was 12% of that in Delay 2 (Component 2).”: Components are conflated with their magnitude.

We have reordered the sentences to avoid confusion:

“The two elements we identified consisted of 7 vectors in the 226-dimensional space, and according to the unmixing coefficients we identified, the magnitude of one element (Element 1) in Delay 2 was 65% of that in Delay 1, and the magnitude of the other element (Element 2) in Delay 1 was 12% of that in Delay 2.”

“The temporal dynamics of the full space population activity projected into these subspaces showed that activity in the first subspace emerged early after target presentation and was maintained until the saccade cue…”: Projections or magnitude of the projections?

We were referring to the magnitude of the projections. We have rewritten the text:

“The temporal dynamics of the full space population activity projected into these subspaces showed that the magnitude of activity in the first subspace increased early after target presentation and was maintained until the saccade cue, while the magnitude of activity in the second subspace increased after distractor presentation and stayed relatively high even after the Go cue (Figure 1C, single-session results are shown in Figure 2—figure supplement 2).”

In what sense is the term “information” used throughout the manuscript? (In the vernacular sense or as a measure of uncertainty in bits?)

We used the term “information” in a general sense throughout the paper, which mostly referred to the neural code enabling the decoding of different intended locations, with the only exception of “mutual information”, which is a measure of uncertainty in bits.

We have added a brief definition of the general “information” in Results:

“Two different stable population activity in LPFC were observed in Delay 1 and Delay 2, which implied that a downstream region would need to use different decoders in the two periods to extract the stable working memory information (neural codes supporting the discrimination of different intended items)…”

“…information emerged right after target presentation, and although the information was stronger in Delay 1 (60.5 ± 1.3%),”: What is the relationship between target information and decoder performance? are both variables measured with a percentage?

Decoder performance is a proxy of target information, and more specifically, it only speaks for linearly decodable information. Only decoder performance is measured with a percentage.

We have added a brief description in the text:

“Next, we used the decoding performance of a linear decoder (LDA) as a proxy of target information and evaluated target information in each subspace.”

“Cross-temporal decoding of full space neural activity projected into the second subspace showed that information emerged after distractor presentation (42.6 ± 1.1%), and was stable throughout Delay 2 (Figure 1F). In Delay 1 and Delay 2, the first subspace explained 14.6% and 10.3% of the full space variance, while the second subspace explained 5.8% and 8.1% of the full space variance, respectively.”: What is the relationship here between the variance explained by subspaces and the components? The distinction between “component” and “subspace” is not always clear.

We have changed the term “Component” into “Element”, and an “Element” refers to a 226 x 7 vector set identified using the optimization method.

“The two elements we identified consisted of 7 vectors in the 226-dimensional space…”

The 7 vectors in an “Element” were not orthogonal to each other, and a “subspace” refers to the orthonormal bases spanned by the vectors from the vector set.

“The orthonormal bases of the two elements defined two subspaces (Subspace 1 and Subspace 2).”

“…and the first 6 out of the 7 principal components cumulatively accounted…”: In standard dimensionality reduction, for example, the goal is to reduce the dimensionality of the neuron state space (226 neurons) to a lower number (e.g., Santhanam et al., 2009). What do the 6 effective dimensions refer to?

From the optimization, we identified two elements, each consisted of 7 226-dimensional vectors. The subspace spanned by each vector set is surely 7 dimensional, given that all the 7 vectors are independent. However, this does not guarantee that the projection of full space data into this subspace will still be 7 dimensional – it is possible that the projection spans an even smaller dimension in the subspace. Hence, we performed a PCA after projecting full space data into the subspace, and the “6 effective dimensions” were referring to the 6 PCA components that explained more than 95% variance. In other words, the “6 effective dimensions” were describing the data projected into the subspace, rather than the subspace per se. We have modified our text to clarify this point.

“Full-space data in the two subspaces had an effective dimensionality of 6 dimensions each – after projecting single-trial full space data into the subspaces, we performed a PCA on the projected data, and the first 6 out of the 7 principal components cumulatively accounted for more than 95% of the variance within each subspace (Figure 2—figure supplement 3).”

Figure 2—figure supplement 3 title:

“Effective dimension of full-space data in the subspaces.”

Subsection “Information in one subspace led to a small amount of interference in information in the other subspace”, “We found no evidence of a drop in performance between proj_MSub_(M) and proj_MSub_(M+P) (p > 0.73, g = 0.61), and between proj_PSub_(P) and proj_PSub_(M+P) (p > 0.22, g = 2.63), suggesting a lack of interference between these subspaces”: How is this result interpreted in light of the finding that the subspaces are not orthogonal?

Because we used LDA’s decoding performance as a proxy of target information, the lack of interference between two non-orthogonal subspaces indicated that the shift of clusters in the state space were not large enough to cross the classification boundaries, and thus did not affect the classification performance. These cases can be observed in Figure 2F and Figure 2—figure supplement 4.

We have more clearly stated this interpretation in the main text.

“As we used LDA’s decoding performance as a proxy of target information, the lack of interference between two non-orthogonal subspaces indicated that the shift of clusters in the state space caused by superimposed activity were not large enough to cross the classification boundaries, and thus did not affect the classification performance.”

b) Novelty and relationship to established methods:Subsection “Two minimally dependent subspaces coexisted within the LPFC”: There is a discussion of the difficulty of obtaining labels for working memory and motor preparation subspaces, and the subsequent application of a novel method to obtain them. How is this different from orthogonalization? In other words, how is considering mutual information between components a more general framework?

We discussed an alternative orthogonalization method in the Discussion.

“In order to assess whether imposing orthogonality between subspaces was feasible, we fixed Delay 1 activity as the first activity subspace and rotated each column vector in Delay 2 activity matrix to be orthogonal to Delay 1 activity (Gram-Schmidt algorithm) to obtain the second activity matrix, such that the two activity matrices would be orthogonal to each other.”

The limitation of the orthogonalization method is discussed further in Figure 5—figure supplement 1.

“First, imposing orthogonality between subspaces, while possible, may hide interesting properties in the data, since activity subspaces could be perfectly orthogonal, but they could also be non-orthogonal, such that interference between them was possible (which may account for interference between cognitive processes). As such, imposing orthogonality would prevent us from identifying interference between subspaces. Instead, the unmixing method allows for both possibilities, and hence is a more unbiased way to understand our data. Second, the unmixing method has fewer assumptions and is more flexible for subspace identification. Orthogonal decomposition imposes one fixed subspace to begin with, and the second subspace is entirely contingent to the blind choice of the first subspace. Instead, the unmixing method simultaneously identifies two subspaces without biasing towards either one.”

Aforementioned subsection, “Our objective was then to find through an optimization technique the best unmixing matrix, to apply to Delay 1 and Delay 2 activity, that could recover the working memory and motor preparation activity with the lowest mutual information possible between them (see Materials and methods).”: How is this different from the optimization method used in Parathasurathy et al., 2019? Is this a general framework to identify any pair of subspaces?

It is true that both methods are under the optimization framework, but their objectives are quite different.

The 2019 paper aimed to find a single subspace in which the projection of Delay 1 and Delay 2 activity overlapped, and the objective function to be minimized was the distance between the Delay 1 and Delay 2 clusters projected into the subspace.

This paper aimed to find two subspaces that were minimally dependent on each other, and the objective function to be minimized was the mutual information between the two unmixed elements.

Given good evidence that there are two distinct subspaces in the data, our method does provide a general framework to identify the subspaces.

“This was qualitatively consistent with our hypothesis, aside from the decrease in information in Delay 2 (39.9 ± 1.1%).”: Is the decrease in “information” in Delay 2 related to the fact that this subspace is morphed in Delay 2 (Parathasarathy et al., 2017)?

Yes, the decrease in decoding performance in the working memory subspace was related to code morphing. This is because in Delay 2, the motor preparation code was superimposed onto the working memory code (which resulted in code-morphing); in the presence of divisive normalization, working memory activity was suppressed, and thus the decoding performance also decreased. This observation was replicated in the bump attractor model with normalization.

Figure 4—figure supplement 2:

“e, Cross-temporal decoding performance of the model (with normalization) in the working memory subspace. Decoding performance reduced significantly in the working memory subspace in Delay 2 (84.1 ± 7.7% in LP11, 58.4 ± 4.1% in LP22, P < 0.05, g = 3.39).”

“Our results provided the first evidence showing that information about two separate cognitive processes can be simultaneously encoded in subspaces within the same brain region.”: This is a strong statement for at least three reasons: (a) there are other studies that have shown some decomposition related to what can arguably be called cognitive processes, including some from the rodent literature which were not referenced (e.g., Inagaki et al., 2019, Svoboda and Li, 2018); (b) this work has not conclusively/causally shown that these subspaces indeed contain information about memory and motor preparation, as lesion studies would; (c) the word subspace in this study does not imply reduced dimensionality as in many other studies (e.g. Kaufman et al., 2014). I suggest rewriting in a more modest form.

We acknowledge the limitations in causality evidence as pointed out in (b) by the reviewer, so we softened our gap statement as:

“Our results provided the first evidence *suggesting* that information about two separate cognitive processes can be simultaneously encoded in subspaces within the same brain region.”

The two rodent studies the reviewer mentioned in (a) (Inagaki et al., 2019, Svoboda and Li ,2018) only explored premotor activity and the premotor subspace, which is still different from our results showing two simultaneous cognitive subspaces (working memory *and* motor preparation). We have added references to them, together with other work showing the existence of one subspace.

“The majority of the literature on information subspaces in the brain has reported a single subspace (Druckmann and Chklovskii, 2012; Inagaki et al., 2019; Parthasarathy et al., 2019; Svoboda and Li, 2018)…”

Additionally, we have added a reference to another recent human study (Minxha et al., 2020), where they reported two cognitive subspaces for a “memory” task and a “categorization” task. However, the two subspaces were employed in different types of trial blocks in an “alternating” fashion, still different from our simultaneous activation of both working memory and motor preparation subspaces.

“Minxha et al., 2020, reported the existence of two cognitive subspaces for “memory” and “categorization” tasks, but the two subspaces were employed asynchronously in different trial blocks.”

The 7-dimensional working memory subspace were identified from experimental data containing 7 target locations, but the real encoding subspace for working memory could require fewer dimensions than 7. For Point (c), we think the fact that the effective dimensions of the data in the working memory subspace was 6, rather than 7, indicated that there was a reduction in dimensionality. We further believe that if there were more task conditions (24 target locations for example), the reduction of effective dimension of data in the subspace will be more prominent, as it will asymptote to the true dimensionality of the working memory code. We have added a few lines after the presentation of “effective dimension” to elaborate on this point.

“Full-space data in the two subspaces had an effective dimensionality of 6 dimensions each … This indicated that the true dimensionality of the neural code could be smaller than the number of discrete target locations imposed by the experiment. In addition, as the number of discrete target locations increases in the experiment (for example, 24 target locations), we expect that the effective dimensionality of data in the subspaces will asymptote to the true dimensionality of the neural codes supporting the cognitive processes.”

c) Overall clarity in description:A method is described to obtain working memory and motor preparation subspaces. However, the logical presentation is not adequate. In Figure 1, for example, subspaces are already introduced to be related to working memory and motor preparation, without any evidence. An alternative is to name the two subspaces in Figure 1 only after showing that they correspond to functional subspaces (before they are, Subspace 1 and Subspace 2).

We agree with the reviewer and have renamed the subspaces as Subspace 1 and 2 before the presentation of evidence.

“Since the second subspace contained target information only after the distractor disappeared, and motor preparation presumably began after the last sensory cue that reliably predicted the timing of the Go cue (i.e. the offset of the distractor), we hypothesized that the second subspace corresponded to a motor preparation subspace”: Can motor preparation and working memory be distinguished in terms of behavior during error trials (e.g., Inagaki et al., 2018)?

Unfortunately, we do not think we can distinguish them with error trials. This is because in correct trials, the working memory location was always the same as the motor preparation location. When an error was made, we would not know if it was due only to incorrect preparation, or due to an incorrect memory plus an incorrect preparation (our error trial analysis suggested that it was likely to be the latter case, as the decoding performance using error trial data decreased in both subspaces, Figure 5B,C). This analysis will only be possible in another experimental setup where the animals’ behavioral responses dissociated memory and saccade locations, and could report both labels separately in a trial.

Additionally, we did not have enough error trials for this analysis. For example, if we looked for trials where the correct location was A, but an incorrect saccade was made to Location B, the number of such trials were no more than 9 among all (A, B) pairs in any single session, making it hard to reach any statistical conclusions.

“In Delay 1 and Delay 2, the first subspace explained 14.6% and 10.3% of the full space variance, while the second subspace explained 5.8% and 8.1% of the full space variance, respectively.”: This sentence about explained variance is difficult to parse, because of the existence of two delays, two subspaces and the full space. What would be the expected total variance? Are the subspaces explaining a fraction of the variance of the full space?

Yes, the subspaces explained a fraction of the variance in the full space. The variance explained by the subspaces were analogous to the variance explained by principal components, and the total variance would be 100% of the variance in the full space. We have rewritten the sentence for better readability.

“In Delay 1, *Subspace 1* and *Subspace 2* accounted for 14.6% and 10.3% of the variance in the full space; in Delay 2, *Subspace 1* and *Subspace 2* accounted for 5.8% and 8.1% of the variance in the full space.”

2) Modelinga) Model construction:i) Equation in subsection “Model” describes the dynamics of the “bump attractor” model. A “leak” term, which reflects a decay towards a baseline firing rate, seems to be missing. From a modeling point of view, the leak is crucial for the dynamics, as its existence is what motivated recurrent synaptic circuitry to counterbalance the decay mediated by the leak for integration and working memory computations as mediated by “bump attractor” dynamics (e.g., compare to Wimmer et al., 2014, which was cited).

The “leak” term was technically not missing in the previous version of our model, it was just hidden. With a “leak” term, the dynamics defining the model would look like: τdrdt=−r+φ(Wrecr+WinI+σ) where r is the firing rate of the population; Wrecr+WinI+σ represents the sum of the recurrent input, external input, and noise; τ is the decaying time constant. To enable numerical simulation, we use Newton’s method to discretize the equation: rt+1=rt+(−rt+φ(Wrecrt+WinI+σ))×dtτ where dt is the length of the simulation time step. In the previous version of the model, we made dt=τ, and the “leak” term was canceled out, so the dynamics looked like: rt+1=φ(Wrecrt+WinI+σ) Now we have set τ=20ms, and dt=2ms, which essentially makes the approximation steps 10 times finer than the previous simulation. We have verified that this change does not result in any qualitative changes to the model results, and have updated the model description and results to the revised version.

“The firing rate of the population was characterized by:

τdrdt=−r+φ(Wrecr+WinI+σ) where τ was a uniform decay constant; . . . For numerical simulation, we used Newton’s method: rt+1=rt+(−rt+φ(Wrecrt+WinI+σ))×dtτrt=rt/αt where we set τ=20ms and dt=2ms; αt was a scalar obtained by mean(rt)/mean(r0), and it was applied uniformly to each unit of the whole population to maintain the mean population firing rate at the baseline level (divisive normalization)…”

ii) There are no details of how “normalization” was implemented. Was it applied to all units uniformly? What is the form of the normalization equation and what are the values of the parameters? It seems like an ad-hoc step, while some models have shown that normalization arises naturally in some implementations of the bump attractor model (e.g., Ardid et al., 2007).

The divisive normalization step in the model was inspired by an observation in the neural data – the mean population firing rate in the delay periods was not different from that in the baseline period. This was different from the normalization described in Ardid et al., 2007, because in that paper, the mean population firing rates before stimulus presentation and during the mnemonic delay were different.

The divisive normalization was applied uniformly to all the units. We divided the firing rate of each neuron by the same factor (derived from the sum of the population activity) to maintain the overall population activity at a constant level. We have added more details to the description of the divisive normalization:

“rt+1=rt/αt…αt was a scalar obtained by mean(rt)/mean(r0), and it was applied uniformly to all units to maintain the mean population firing rate at the baseline level (divisive normalization).”

We acknowledge that this implementation of normalization simulated only the high-level operation of divisive normalization; more biological-plausible mechanisms and detailed modeling on the single-neuron level could be explored for future work.

iii) Subsection “Model” paragraph two: The description of the inputs here (i.e., working memory and motor preparation) is missing some details. Are the inputs correlated? How does the distractor input affect the overall dynamics? What are their values?

Yes, the working memory input and the motor preparation were correlated (1-to-1 mapping) in the model. In each trial, the distractor location was always different from the target location; the input loadings of the distractor was the same as that of the working memory input, but the strength was only 50%. In other words, working memory input was provided during target presentation period, and distractor input and motor preparation input were provided during distractor presentation period. We have added the following description:

“In each trial, the target label for working memory and motor preparation input was always the same. Distractors used the same input loadings as the working memory input, but the strength was only 50% and the distractor label was always different from the target label.”

We have added a new analysis to show that distractor activity does not relate to the second subspace (Figure 3—figure supplement 3) or drive code-morphing in bump attractor models (Parthasarathy et al., 2019). Stronger distractor input would decrease the Delay 2 decoding performance as it increased the within-cluster variance if data was grouped by target labels. We have added a new panel (Panel J) to Figure 4—figure supplement 2 to show the relationship between Delay 2 performance and strength of distractor input.

iv) In subsection “A bump attractor artificial neural network with divisive normalization recapitulated the properties of LPFC activity” there is a description of how the model is constrained by selectivity of the neural data, as well as by the “43 % overlap” in inputs (Figure 5). How this related to the loadings found in Figure 3?

We reported that in the neural data, 43% of the neurons exhibited selectivity to both working memory and motor preparation (Figure 4—figure supplement 3), and the “43% overlap” in the inputs in the model replicated this observation: 43% of the units in the model received both working memory and motor preparation inputs, and hence were selective to both.

Loading and selectivity are slightly different. Loading measures how a neuron’s activity aligns to a subspace, but it does not guarantee selectivity in the subspace; selectivity is only provided when the responses of the neuron are different for different stimuli. We only constrained the model by selectivity ratio, not loadings for the subspaces.

b) Model significance:i) What are the activities of the model neurons? In particular, it is not clear whether the neurons are mixed selective, i.e., whether any given (simulated) neuron participates in both working memory and motor preparation. A hint is given in Figure 4—figure supplement 2, but this should be expanded.

We have added diagrams showing single-unit activity in the model in Figure 4—figure supplement 2G.

ii) In Figure 5, a direct comparison between model and data is missing, although it is described qualitatively in the text.

We compared the bump attractor model with data quantitatively in Figure 4—figure supplement 2. Now we have also added quantitative comparisons between the linear subspace model and data in Figure 4—figure supplement 4.

“…b, Cross-temporal decoding of the model (without normalization) in the full space. Delay 2 decoding performance (86.9 ± 1.3%) was significantly higher than Delay 1 performance (65.8 ± 1.2%). This was inconsistent with our observations from the neural data. c, Cross-temporal decoding of the model (with normalization) in the full space. Code morphing was replicated in the full space; Delay 2 decoding performance (65.0 ± 1.5%) was not significantly different from Delay 1 performance (65.0 ± 2.0%). d, Cross-temporal decoding performance of the model (with normalization) in the working memory subspace. The decay of working memory information was replicated in the working memory subspace identified by the unmixing method; decoding performance reduced significantly in the working memory subspace in Delay 2 (81.5 ± 1.0% in LP_11_, 58.8 ± 2.6% in LP_22_, P < 0.01, g = 11.3). e, Cross-temporal decoding performance of the model (with normalization) in the motor preparation subspace. As expected, target information in the motor preparation subspace emerged in Delay 2…”

iii) What are the model predictions?

If the two ring attractor networks for working memory and motor preparation indeed overlaps as the model assumes, then it predicts that if we sort the neurons according to Delay 1 activity to form a ring structure, we would expect new “bumps” representing motor preparation activity in Delay 2. We are currently conducting new analysis on more neural data to test this hypothesis. We have added this prediction to Figure 4—figure supplement 2.

“This architecture predicts that if we sort neurons according to the working memory “bumps” in Delay 1, we would be able to see the “bumps” representing motor preparation in Delay 2.”

3) Additional analyses or plots to improve claritya) Is Figure 1C necessary, given a similar Figure 1G with actual data? A general schematic is always welcome, perhaps illustrating distinct possibilities of overlap between the subspaces.

(The two plots are now Figure 2B and Figure 2F.)

Figure 2B is different from Figure 2F, in that it is a more intuitive prediction of the dynamics in the subspaces. One important aspect in Figure 2F, which the schematic (Figure 2B) failed to predict, is the divisive normalization of activity. We have highlighted the difference due to divisive normalization between Figure 2B and Figure 2F:

“The reduction of working memory decoding performance was not expected in the schematic diagram of the subspace dynamics (Figure 2B), but was captured by the state space visualization of real neural data (Figure 2F, inter-cluster distance in *Subspace 1* reduced in Delay 2).”

b) “In an additional test of the hypothesis that the second subspace corresponded to a motor preparation subspace, we examined the relationship between the unmixed motor preparation activity and the unmixed pre-saccade activity at the level of single cells”: This is a continuation, at a single cell level, of the previous hypothesis of relating delay 2 period activity to pre-saccadic activity. For an independent analysis, can the activity in error trials be used to distinguish motor preparation from working memory representations?

Unfortunately, we do not think we can distinguish them in error trials with our data. Details are discussed in point c.

c) “Importantly, Component 2 and Component 2’ were also significantly correlated (Figure 2A right, Pearson correlation r = 0.62, P <0.01).”: Can the similarity between the components be defined in terms of angles, as done between the subspaces?

(We have now renamed a “component” as an “element”, which is a 226 x 7 vector set.)

The two subspaces are the orthonormal bases of the two elements, and because the principal angles between two elements are essentially the principal angles between their orthonormal bases, such an analysis is the same as in Figure 5A.

Moreover, the correlation component2-component2' = 0.62 is high, but could be higher. How does the motor preparation subspace relate to distractor encoding? Could this explain why the correlation 2-2' is not higher?

The second subspace is not related to distractor encoding (see response to question 3 of reviewer #1). We have added a new section and a supplementary figure to address this point.

“Alongside the working memory and motor preparation activities for target locations, there could also be activities representing distractor locations in Delay 2. By grouping trials according to distractor labels, we indeed found significant distractor information in the full space (Figure 3—figure supplement 3). However, the distractor activity in Delay 2 was not related to the *Element 2* or the motor preparation subspace we identified, because the distractor activity and the motor preparation activity were obtained from data grouped by different trial labels (target and distractor labels were uncorrelated). Very little distractor information (17.9 ± 0.7%) was successfully decoded in the motor preparation subspace (Figure 3—figure supplement 3).”

And see the new Figure 3—figure supplement 3.

Even though motor preparation and pre-saccade activity are correlated, we do not expect them to be identical, as there could be other transformations between preparatory and execution activities (Churchland et al., 2012; Kaufman et al., 2014). We believe this should be the main reason why the correlation between 2-2’ is not higher.

d) “…concurrent increase of activity in Neurons 1 and 2 signals a change in memory (i.e. the population activity in state space moved along the working memory subspace)…”: This has not been shown explicitly with the neural activity, namely, that moving along the working memory subspace, corresponds to a (discrete) change in the target representation. Such figure would be much more informative than the illustration in Figure 3C (which is appreciated, but would actually be valid for any orthogonal subspace decomposition, not necessarily working memory/motor preparation).

We agree with the reviewer that real neural activity would be more informative than a schematic, and we indeed showed the low-dimensional state space visualization of the neural activity in the working memory/motor preparation subspaces (Figure 2F, Figure 2—figure supplement 4), which should be more informative than showing single-unit activity. We have changed our text to refer to these two figures for neural activity visualization.

“This concept can be extended to the 212 neurons with mixed selectivity to understand how the coordinated activity between those neurons can contribute minimally dependent information to the working memory and motor preparation subspaces through different loading weights that we found in the LPFC (low-dimensional visualizations of neural data provided in Figure 2F and Figure 2—figure supplement 4).”

The purpose of this schematic is more than a visualization of population activity. What we wanted to emphasize was the “readout” aspect of the population coding – how could the subspaces with specific loading weights read out different and minimally dependent information from a single population of mixed-selective neurons. We have changed the figure and text to elaborate on this point.

“In order to understand how a single population of neurons with mixed selectivity could have contributed minimally dependent information to the two subspaces, we created a simple illustration (Figure 4C). Working memory and motor preparation information were read out by separate readout neurons with different connection weights to Neurons 1 and 2 that reflected the loading weights of each subspace. In isolation, the activity of *Neuron 1* would be ambiguous for both readout neurons, as an increase of activity in Delay 2 could be interpreted as a new memory at a different spatial location, or as the same memory as in Delay 1, but with superimposed motor preparation activity. In order to disambiguate the meaning of a change in the activity of one neuron, it would be necessary to interpret that change in the context of changes in the activity of the rest of the neuronal population (i.e. in this example, *Neuron 2*). In the illustration, a superimposed increase of activity in Neurons 1 and 2 signals a change in memory (i.e. only the readout activity in the working memory subspace changed), whereas the same increase in *Neuron 1*, but with a superimposed decrease of activity in *Neuron 2*, signals that the memory has not changed, but that a motor preparation plan has emerged in Delay 2 (i.e. only the readout activity in the motor preparation subspace changed).”